# An architectural role of specific RNA–RNA interactions in *oskar* granules

Mainak Bose[1,4,6], Branislava Rankovic [1,5,6], Julia Mahamid [2,3] ✉ & Anne Ephrussi [1] ✉

Ribonucleoprotein (RNP) granules are membraneless condensates that organize the intracellular space by compartmentalization of specific RNAs and proteins. Studies have shown that RNA tunes the phase behaviour of RNA-binding proteins, but the role of intermolecular RNA–RNA interactions in RNP granules in vivo remains less explored. Here we determine the role of a sequence-specific RNA–RNA kissing-loop interaction in assembly of mesoscale *oskar* RNP granules in the female *Drosophila* germline. We show that a two-nucleotide mutation that disrupts kissing-loop-mediated *oskar* messenger RNA dimerization impairs condensate formation in vitro and *oskar* granule assembly in the developing oocyte, leading to defective posterior localization of the RNA and abrogation of *oskar*-associated processing bodies upon nutritional stress. This specific *trans* RNA–RNA interaction acts synergistically with the scaffold RNA-binding protein, Bruno, in driving condensate assembly. Our study highlights the architectural contribution of an mRNA and its specific secondary structure and tertiary interactions to the formation of an RNP granule that is essential for embryonic development.

Eukaryotic cells contain a large number of RNA–protein condensates, broadly known as RNP granules, including stress granules, germ granules, neuronal transport granules and others[1]. These membraneless compartments are enriched in RNAs and RNA-binding proteins (RBPs), many of which harbour intrinsically disordered regions (IDRs) and prion-like domains (PrLDs)[2–4]. Recent studies on biomolecular condensation have highlighted the role of RNA–RBP interactions in regulating the assembly of RNP granules[1,3,4]. Here, the synergistic action of stereospecific RNA–RBP and multivalent IDR–IDR interactions drives the formation of multi-component condensates, which have also been described using the theoretical 'scaffold–client' framework[5,6]. A generic model of RNP granule formation postulates that RNA binding can promote high local concentration of IDR-containing RBPs, which, through protein–protein interactions, connect individual RNP complexes into mesoscale condensates. What is commonly neglected in such models is the potential role of intermolecular RNA–RNA interactions. Long RNA molecules can form higher-order assemblies also in the absence of proteins[7]. This is exemplified by protein-free condensation of RNA homopolymers in vitro[8], cell-free total yeast RNA[9] and pathogenic repeat expansion RNAs[10]. The nature and biophysical properties of intermolecular RNA–RNA interactions span a continuum from high-affinity sequence-specific interactions to promiscuous base pairing between exposed sequences along large RNAs[1,11].

*oskar* granules in the *Drosophila melanogaster* female germline are a class of transport RNPs that package and localize the maternal RNA *oskar* to the posterior of the developing oocyte. The locally translated Oskar protein is essential for abdominal patterning and specification of germ cell fate during embryonic development[12,13]. We have recently reported that *oskar* granules are phase-separated condensates with solid-like material properties[14]. Using a combination of in vitro and

[1]Developmental Biology Unit, European Molecular Biology Laboratory, Heidelberg, Germany. [2]Structural and Computational Biology Unit, European Molecular Biology Laboratory, Heidelberg, Germany. [3]Cell Biology and Biophysics Unit, European Molecular Biology Laboratory, Heidelberg, Germany. [4]Present address: Department of Bioscience and Biotechnology, Indian Institute of Technology, Kharagpur, India. [5]Present address: Laboratory of Molecular Neuroscience, German Center for Neurodegenerative Diseases, Berlin, Germany. [6]These authors contributed equally: Mainak Bose, Branislava Rankovic. ✉e-mail: julia.mahamid@embl.de; anne.ephrussi@embl.org

in vivo assays, we identified the RBP Bruno as a primary scaffold protein that is crucial for *oskar* granule formation and their liquid-to-solid phase transition. The RNA recognition motifs of Bruno bind-specific sequences (Bruno response elements, BREs) in the *oskar* 3′ untranslated region (UTR)[15]. Bruno N-terminal PrLD self-association drives assembly of mesoscale *oskar* granules, which partition client proteins in an RNA-dependent manner to regulate diverse aspects of *oskar* function, such as translation regulation[14]. Our fluorescence microscopy-based quantifications estimated an average of 16 *oskar* messenger RNA (3 kb) molecules packaged in ~400-nm-diameter condensates, amounting to an RNA concentration of 873 nM (ref. 14). The high RNA concentration suggests a potential architectural role of *oskar* mRNA in granule assembly or organization. In fact, our minimal in vitro reconstitutions with *oskar* and Bruno indicated formation and stabilization of RNA–RNA interactions upon Bruno-driven condensation[14]. However, the nature of *trans* RNA–RNA interactions and their contribution to *oskar* granule condensation remained to be explored.

## Results

### RNA kissing-loop interaction is essential for *oskar* granules

The 3′ UTR of *oskar* mRNA harbours a 67-nucleotide-long stem-loop structure, SL2b, also referred to as the oocyte entry signal (OES) (Fig. 1a). The AU-rich stem of SL2b serves as a *cis*-acting RNA localization signal essential for dynein-dependent transport of *oskar* from the nurse cells to the oocyte[16]. The terminal loop of the same stem-loop structure harbours a six-nucleotide palindromic sequence (5′-CCGCGG-3′) known to promote dimerization of *oskar* mRNA in vitro via a kissing-loop interaction through canonical Watson–Crick base pairing[17] (Fig. 1b). Dimerization of the OES is robust in vitro and occurs in absence of any monovalent or divalent cations in the buffer. Aiming to disrupt the kissing-loop intermolecular interactions, we introduced a two-nucleotide substitution (5′-UUGCGG-3′) in the palindrome, hereon referred to as *oskar* UU[17]. This indeed abolished OES dimer formation, even under conditions with high concentrations of Na$^+$ and Mg$^{2+}$ ions (Fig. 1b and Extended Data Fig. 1a). We previously reported that, in vivo, the kissing-loop-based dimerization promotes co-packaging of transgenic reporter RNAs with endogenous *oskar* RNA[17,18]. This observation suggests a role of the kissing loop in assembly of endogenous *oskar* granules. To test this hypothesis, we expressed genomic *oskar* transgenes comprising either a wild type (*oskar* WT) or a mutated dimerization domain (*oskar* UU) in flies in which no endogenous *oskar* RNA is expressed. In absence of *oskar*, oogenesis fails to progress[19,20]. Both the transgenic WT and UU RNAs enriched in the early oocyte and rescued progression of oogenesis in the *oskar* RNA null flies (Fig. 1c). Therefore, the two-nucleotide substitution in the loop does not disrupt the recruitment of the dynein-transport machinery to the OES stem[16], suggesting that the UU mutation does not interfere with the secondary structure of the stem loop. From mid-oogenesis onwards, however, the UU RNA mislocalized along the oocyte cortex and frequently accumulated as a cluster near the posterior pole, whereas the WT RNA robustly localized at the posterior pole of stage 10A egg chambers (Fig. 1c,d). The *oskar* UU transport phenotype was not due to defects in oocyte polarity, as evident from the antero-lateral position of the oocyte nucleus at stage 9, proper localization of *gurken* mRNA and organization of the oocyte microtubule network (Extended Data Fig. 2a,b). Careful examination revealed a diffuse distribution of the *oskar* RNA signal in the *oskar* UU oocytes, in contrast to the distinct, granular signal in the *oskar* WT, suggesting a defect in *oskar* granule formation in the mutant (Fig. 1e). Quantification of the RNA signal showed a significant reduction in partitioning of *oskar* UU RNA into granules, confirming that loss of the kissing-loop interaction interferes with granule assembly in vivo (Fig. 1f). As a consequence, the *oskar* translational regulation was impaired, resulting in ectopic accumulation of Oskar protein in the oocytes and embryos and aberrant embryonic segmentation (Extended Data Fig. 3a,b).

We previously identified Bruno, a translation repressor of *oskar*, as a scaffold protein that drives *oskar* granule assembly in vivo in the presence of WT *oskar* mRNA[14]. The present observations suggest that in addition to Bruno, the *oskar* mRNA itself might play a structural role in scaffolding the granules in vivo.

### The kissing loop acts as a specific RNA 'sticker'

To understand the mechanism of the kissing-loop interaction in scaffolding granules, we truncated the *oskar* 3′ UTR to include the last 359 nucleotides, which harbour the OES and the 3′-most 166 nucleotides of the RNA that are crucial for oogenesis progression[20]. The in vitro transcribed WT$_{359}$ RNA was predominantly dimeric and also oligomerized into multiple higher-order species, as evident from their slower mobility on a native gel in the electrophoretic mobility shift assay (EMSA; Fig. 2a and Extended Data Fig. 1b). In contrast, the UU$_{359}$ RNA exhibited significantly less dimerization and higher-order oligomerization (Fig. 2a,b). Since the mutant OES (UU$_{OES}$) alone failed to dimerize in vitro (Fig. 1b), the observed dimeric form of UU$_{359}$ potentially arises from promiscuous RNA–RNA contacts. Indeed, a dimeric species was still detectable when the entire stem-loop was deleted from the 359 nucleotide long RNA (*oskar*$_{292}$, Fig. 2c), confirming that additional RNA–RNA interactions were promoted by sequences outside the stem-loop structure. However, when subjected to a thermal gradient, the dimeric form of the WT$_{359}$ was more resistant to denaturation than that of the UU$_{359}$ or *oskar*$_{292}$ (Fig. 2c), confirming that in contrast to promiscuous, weak RNA–RNA interactions, the sequence-specific kissing-loop interaction stabilizes the RNA dimer. Notably, the higher-order species observed in WT$_{359}$ persisted in stringent buffer conditions (zero salt), indicating that the WT$_{359}$ RNA can robustly self-assemble by virtue of the kissing loop as well as additional RNA–RNA interactions promoted by the remainder of the RNA sequence (Fig. 2c,d). Upon heating, these higher-order oligomers melted faster than the kissing-loop-induced dimer, further highlighting the strength of the kissing-loop interaction in stabilizing RNA tertiary contacts (Fig. 2c). Therefore, the kissing loop acts as a specific RNA 'sticker' in initiating RNA multimerization, which is propagated by additional non-specific intermolecular RNA 'stickers', while the flanking 'spacer' RNA sequences contribute to expanding the network, potentially by providing binding sites for RBPs[21,22] (Fig. 2d).

### The kissing loop regulates RBP association with *oskar*

We next aimed to examine whether the specific RNA kissing-loop interaction and the consequent RNA assembly play a role in the association of bona fide *oskar* RBPs. Transcript-specific isolation of RNPs demands a large amount of starting material and is experimentally challenging in the case of *Drosophila* oocytes[23]. Therefore, we resorted to an ex vivo approach, whereby we biotinylated the synthesized WT$_{359}$/UU$_{359}$ RNA at the 3′ end, immobilized the RNA on magnetic streptavidin beads and incubated it with ovary extracts from WT flies (Fig. 3a). Bona fide *oskar* RBPs such as Bruno[15], Staufen[24] and polypyrimidine tract-binding protein (PTB)[25] were specifically captured by both WT$_{359}$ and UU$_{359}$ using this affinity pull down method. However, the amount of the core scaffold protein Bruno associated with UU$_{359}$ was significantly lower than with the WT$_{359}$ RNA (Fig. 3a and Extended Data Fig. 4a,b). Since both the WT$_{359}$ and UU$_{359}$ RNAs contain the BRE C, where Bruno is expected to bind, differential recruitment of Bruno suggests a potential role of RNA dimerization and higher-order oligomerization on Bruno association.

To investigate if Bruno is also differentially associated with the *oskar* WT and UU RNAs in the egg chamber, we used Bruno-enhanced green fluorescent protein (EGFP) knock-in (KI) flies[26] in which the WT and UU transgenes were expressed in absence of endogenous *oskar* RNA. In the cytoplasm of the nurse cells, Bruno–EGFP signal co-localized with *oskar* RNA on microtubule tracks[14] for both *oskar* WT and *oskar* UU (Fig. 3b), indicating that the UU mutation does not abrogate Bruno binding in vivo. However, in the oocyte, as opposed to the granular signal and strong co-localization observed with *oskar*

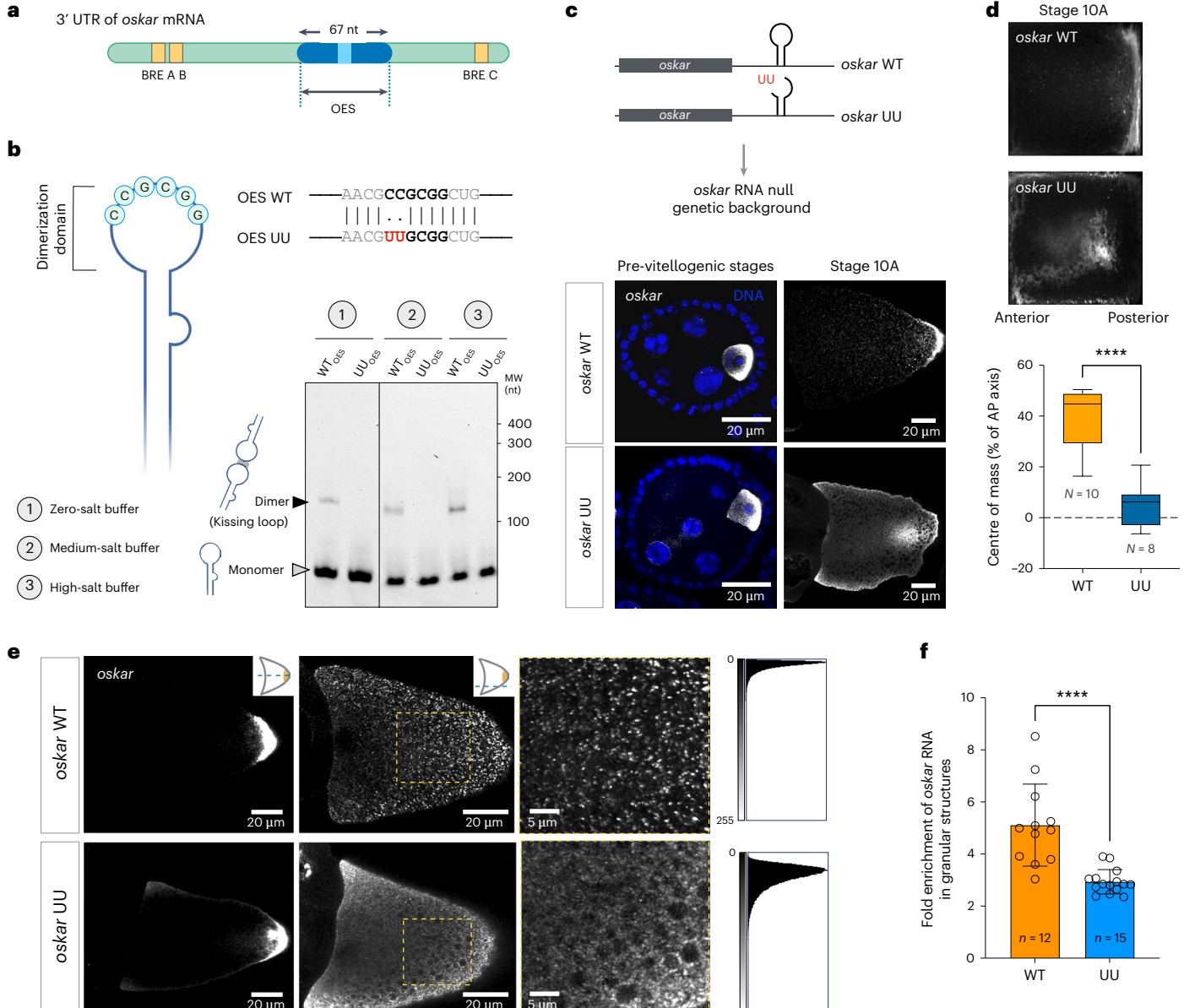

**Fig. 1 | A kissing-loop RNA–RNA interaction is essential for *oskar* granule formation in vivo. a**, A schematic representation of the *oskar* 3′ UTR showing the relative positions of the OES and the BREs. **b**, A cartoon representation of the 67-nucleotide-long OES highlighting the palindromic sequence that engages in a kissing-loop interaction with another *oskar* molecule and dimerization of the WT and UU mutant OES in buffers with varying salt concentrations using atto633-labelled in vitro transcribed RNAs. MW, molecular weight marker. **c**, Localization of *oskar* mRNA (greyscale) in pre-vitellogenic and stage 10 egg chambers detected by smFISH in *oskar* WT and *oskar* UU transgenic flies. Experiments were carried out in flies genetically null for endogenous *oskar* RNA. **d**, Average *oskar* RNA signal (greyscale) from multiple stage 9–10 oocytes, form anterior to posterior. The quantile boxplots display data from the 25th to 75th percentile, median and whiskers extending to the minimum and maximum values of the data set. The dotted horizontal line shows the position of the *oskar* centre of

mass relative to the geometric centre of the oocyte along the antero-posterior (AP) axis. $n = 10$ and 8 oocytes from *oskar* WT and *oskar* UU flies, respectively, from three separate experiments. Unpaired two-tailed Student's *t*-tests were used for comparisons. ****$P < 0.0001$. **e**, Representative confocal images of *oskar* RNA (greyscale) in the *oskar* WT and *oskar* UU oocytes at the equatorial (left) and cortical (middle) planes of the egg chamber. Right: enlarged views of the cortical plane images (dashed yellow box). Histograms of pixel intensities of the enlarged area reveal the diffuse, non-punctate signal of *oskar* RNA in the case of *oskar* UU. **f**, Intensity-based segmentation of *oskar* puncta from the cytoplasm was performed, and the enrichment of *oskar* RNA in granules was quantified. Unpaired two-tailed Student's *t*-tests were used for comparisons. ****$P < 0.0001$. The data are presented as mean values ± standard deviation, and *n* denotes the number of oocytes analysed. $n = 12$ and 15 oocytes from *oskar* WT and *oskar* UU flies, respectively, from three separate experiments.

---

WT, the Bruno–EGFP signal was largely diffuse and rarely co-localized with *oskar* RNA signal in the case of the UU mutant (Fig. 3b). Together with our ex vivo pulldown results, which show the importance of kissing-loop-driven higher-order RNA oligomerization for Bruno association, these findings suggest that mutation of the kissing loop does not impair binding of the scaffold protein Bruno but hinders

higher-order oligomerization of the Bruno–RNA complex, thus aborting granule assembly in vivo. To test this, we performed EMSAs with Bruno–EGFP and *oskar*. Although both the WT$_{359}$ and UU$_{359}$ RNAs bound Bruno, the WT$_{359}$ formed higher-order RNP complexes to a greater extent than the UU$_{359}$ RNA (Fig. 3c). A similar difference in higher-order complex formation was observed with the full length *oskar* 3′ UTR

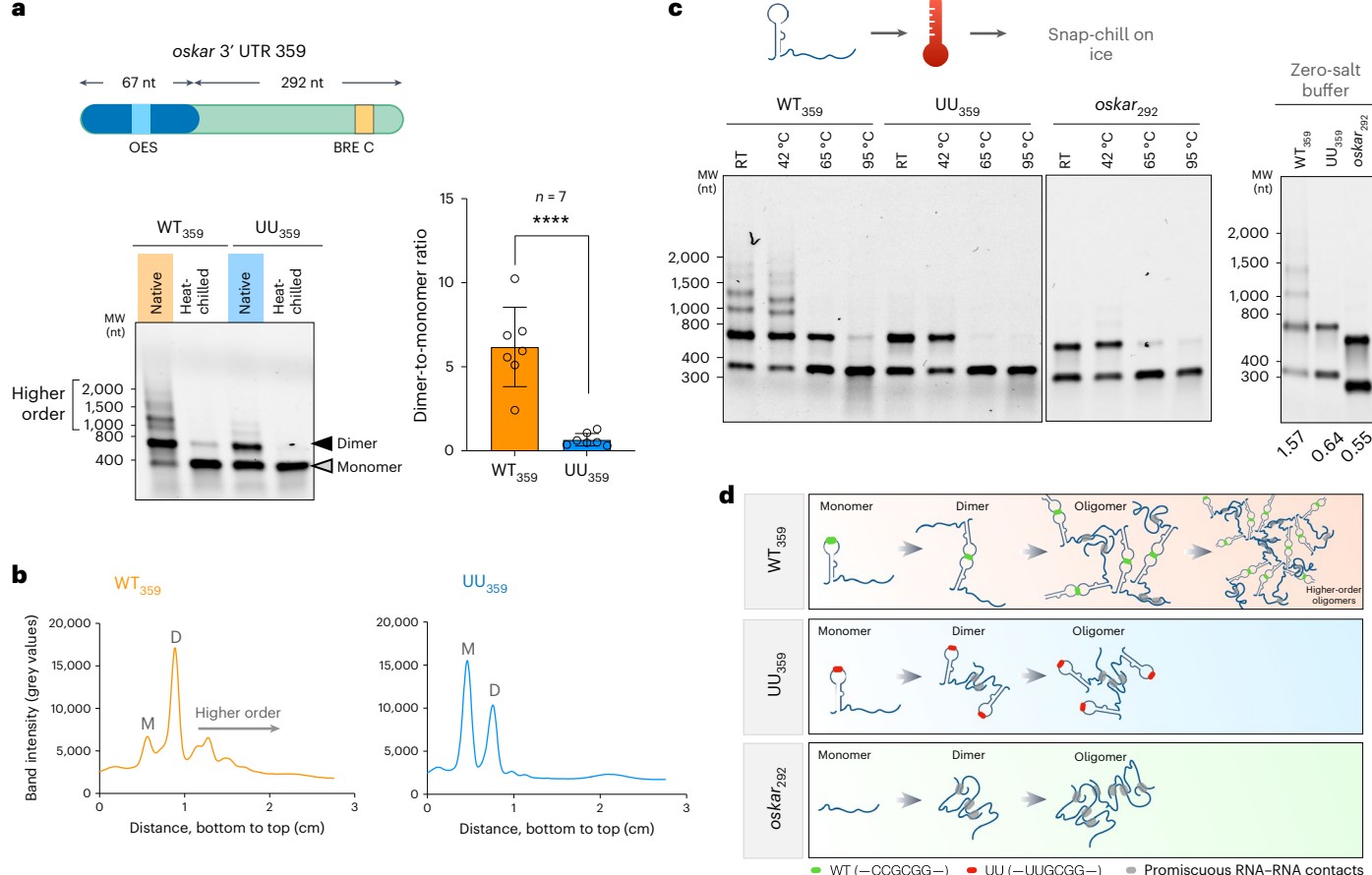

**Fig. 2 | Mutation in the kissing loop impairs RNA dimerization and higher-order assembly. a**, Schematic representation of the 359 nucleotide fragment of *oskar* 3′ UTR; the 67 and 292 nucleotide subfragments are indicated. An electrophoresis of the atto633-labelled WT$_{359}$ and UU$_{359}$ RNAs under native and heat-chilled conditions in EMSA buffer (containing 150 mM NaCl, 2 mM MgCl$_2$) shows enhanced formation of dimeric and higher-order oligomeric species by the WT$_{359}$ RNA. The graph on the right represents the dimer-to-monomer ratio quantified from multiple electrophoresis experiments. The data are presented as mean values ± standard deviation and $n = 7$ independent gel electrophoresis experiments with both conditions run simultaneously each time. ****$P < 0.0001$. Unpaired two-tailed Student's *t*-tests were used for comparisons. **b**, Lane intensity profiles of the RNA signal of WT$_{359}$ and UU$_{359}$ under native conditions (as indicated in **a**). 'D' indicates the dimeric form of the RNAs, and 'M' represents the monomer. **c**, Left: WT$_{359}$, UU$_{359}$ and *oskar*$_{292}$ RNAs were incubated in EMSA buffer at the indicated temperatures followed by snap-chilling on ice and subsequent native agarose gel electrophoresis. Right: behaviour of the WT$_{359}$, UU$_{359}$ and *oskar*$_{292}$ RNA fragments under stringent buffer conditions in absence of Na$^+$ and Mg$^{2+}$ ions shows that both the OES mutant (UU$_{359}$) and the OES-deleted (*oskar*$_{292}$) RNAs form dimers in vitro. The dimer-to-monomer ratio for each condition is denoted below the gel. MW, molecular weight marker. **d**, A graphical model illustrating the transition from monomeric to oligomeric species of the indicated RNAs. Note that, owing to the degenerative nature of RNA–RNA base pairing, promiscuous intermolecular interactions (grey ovals) are prevalent for all three RNAs.

(Extended Data Figs. 1c and 5a,b). EMSA with the OES alone did not show any detectable Bruno binding (Fig. 3d), confirming that the stem-loop does not associate with Bruno and suggesting that Bruno binding is restricted to the remaining 292 nucleotide sequence (*oskar*$_{292}$), which harbours the BRE C. Indeed, the *oskar*$_{292}$ RNA bound Bruno and formed higher-order oligomers, showing that Bruno binding/oligomerization can be uncoupled from RNA dimerization (Fig. 3d). Since the 292 nucleotide sequence is identical in the WT$_{359}$ and UU$_{359}$ RNAs, this observation strongly suggests that in addition to Bruno-driven oligomerization, the kissing-loop interaction contributes to higher-order species formation.

**The kissing-loop interaction drives condensation with Bruno**

Given the established role of Bruno as a granule scaffold[14], we next investigated the significance of RNA–RNA interactions in the context of Bruno-driven condensate assembly in vitro. The experiments were carried out using Bruno below the saturation concentration ($C_{sat}$), where Bruno is soluble and its phase separation is driven only upon addition of RNA. The RNA fragments also did not form any visible assemblies on their own under the chosen close-to-physiological buffer conditions

with 150 mM NaCl and devoid of crowding agents (Fig. 4a). However, mixing the protein and the RNA led to spontaneous co-condensation of Bruno and WT$_{359}$ into microscopic condensates, whereas the UU$_{359}$ and Bruno formed few assemblies that were significantly smaller in size (Fig. 4b,c). Interestingly, the *oskar*$_{292}$ RNA resulted in a reduced condensation similar to the UU$_{359}$ (Fig. 4b,c), despite its binding and oligomerization with Bruno (Fig. 3d). Therefore, the kissing-loop interaction appears to be critical for condensation together with Bruno.

We next sought to investigate whether restoring base pairing of the UU mutant by introducing a compensatory AA mutation[17] in the kissing-loop hexanucleotide could rescue condensation (Extended Data Fig. 6a). The compensatory AA mutation restored in vitro condensation with Bruno only minimally (Extended Data Fig. 6b). We reasoned that the limited rescue could be due to the lower strength of base pairing of A–U, in comparison to the G–C present in the WT *oskar* sequence. To address the effect of the strength of the kissing-loop interaction on condensation, we replaced the WT *oskar* loop sequence with unrelated sequences from two isotypes of the human immunodeficiency virus (HIV) SL1 kissing loop: HIV-1 (GC content of 66.7%)

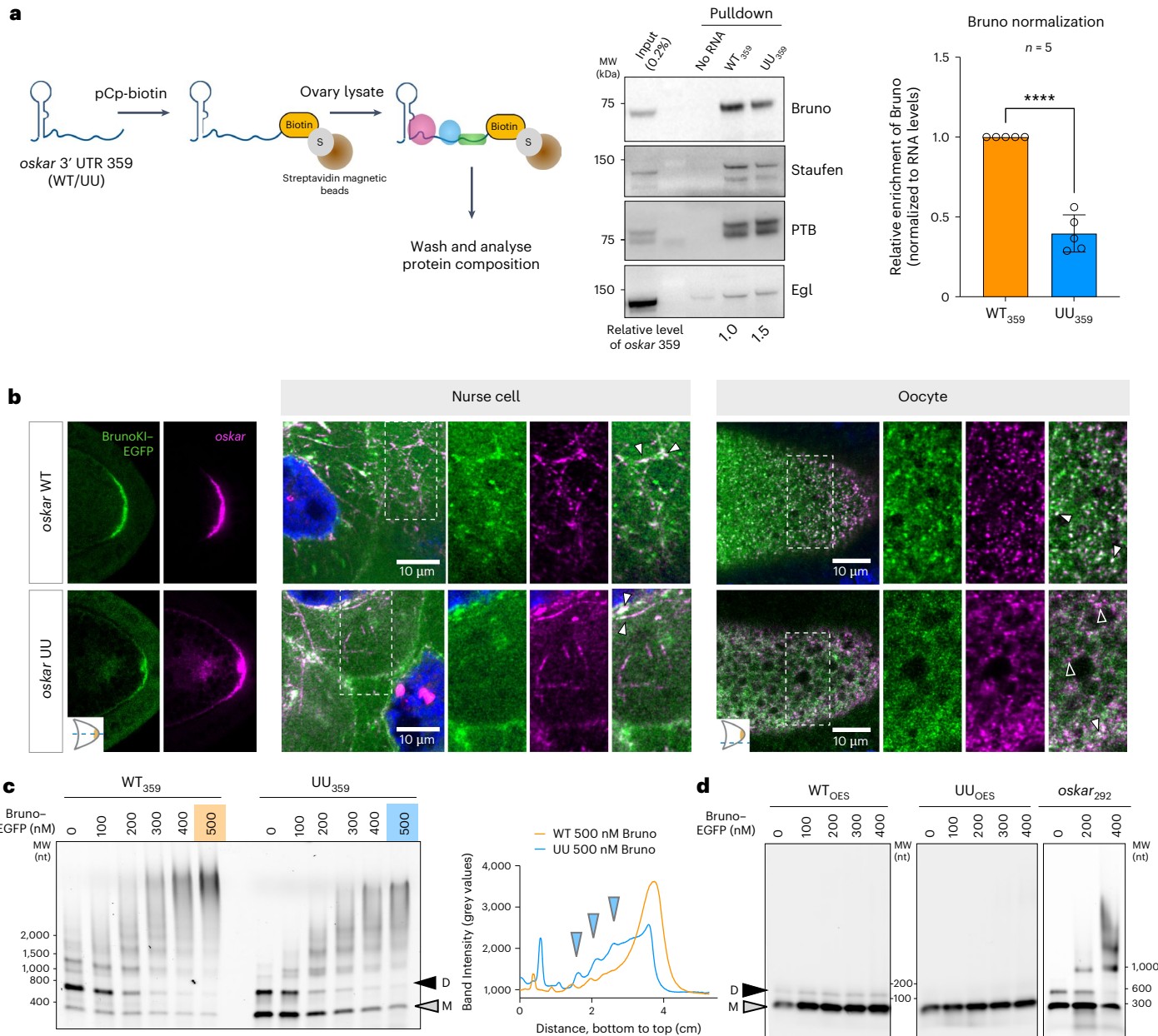

**Fig. 3 | Mutation in the kissing loop reduces association with scaffold protein Bruno. a**, Left: schematic workflow of the RNA affinity capture experiment performed using 3′-biotinylated $WT_{359}$ and $UU_{359}$ RNAs and ovary lysate from WT *Oregon-R* flies. The beads without RNA serve as a negative control. Middle: representative western blot data showing differential association of bona fide *oskar* granule RBPs. Egl, egalitarian. The relative level of the respective RNAs (determined by qPCR) pulled down by the streptavidin beads are indicated below the blot. Right: quantification of the Bruno levels was carried out using western blots from five independent replicates after normalization to the amount of pulled-down RNA. Note that these are not absolute values. The error bar on UU represents the variation in 'relative enrichment' of Bruno between WT and UU in the different experimental replicates. It does not reflect the inherent variation within one sample. The data are presented as mean values ± standard deviation and n = 5 independent biological replicates in which $WT_{359}$ and $UU_{359}$ were analysed in parallel. Unpaired two-tailed Student's *t*-tests were used for comparisons. ****$P < 0.0001$. **b**, Localization of Bruno–EGFP (green) and *oskar* WT

or UU RNA (magenta) in vivo. Left: snapshots of the oocyte posterior (equatorial plane) show enrichment of Bruno KI–EGFP at sites where *oskar* RNA is highly concentrated. Right: 1-μm-thick confocal slice of nurse cell and oocyte (cortical plane); the boxed areas enlarged on the right. The filled white arrowheads represent the co-localization of Bruno and RNA, and the empty arrowheads represent the lack of association of *oskar* RNA signal with Bruno (in *oskar* UU). **c**, EMSA using 50 nM atto633-labelled $WT_{359}$ and $UU_{359}$ RNAs with indicated concentrations of Bruno–EGFP. 'D' indicates the dimeric form of the RNAs, and 'M' represents the monomer. The lane intensity profiles (for the indicated lanes, from the bottom to top of the gel) at the 500 nM Bruno concentration are plotted on the right. The blue arrowheads indicate intermediate oligomeric complexes detectable only in the case of $UU_{359}$. **d**, EMSA using 50 nM atto633-labelled $WT_{OES}$, $UU_{OES}$ and *oskar*$_{292}$ RNAs with indicated concentrations of Bruno–EGFP showing that RNA dimerization can be uncoupled from Bruno-driven oligomerization. MW, molecular weight marker.

and HIV-2 (GC content of 100%)[27–29]. The HIV-2 sequence, with higher GC content, was predominantly dimeric and formed higher-order oligomers, whereas the HIV-1 sequence, with lower GC content, was

visibly less dimeric and failed to form higher-order oligomers (Fig. 4d). Addition of Bruno led to significantly more condensate formation in the case of the HIV-2 RNA compared with HIV-1, suggesting that stronger

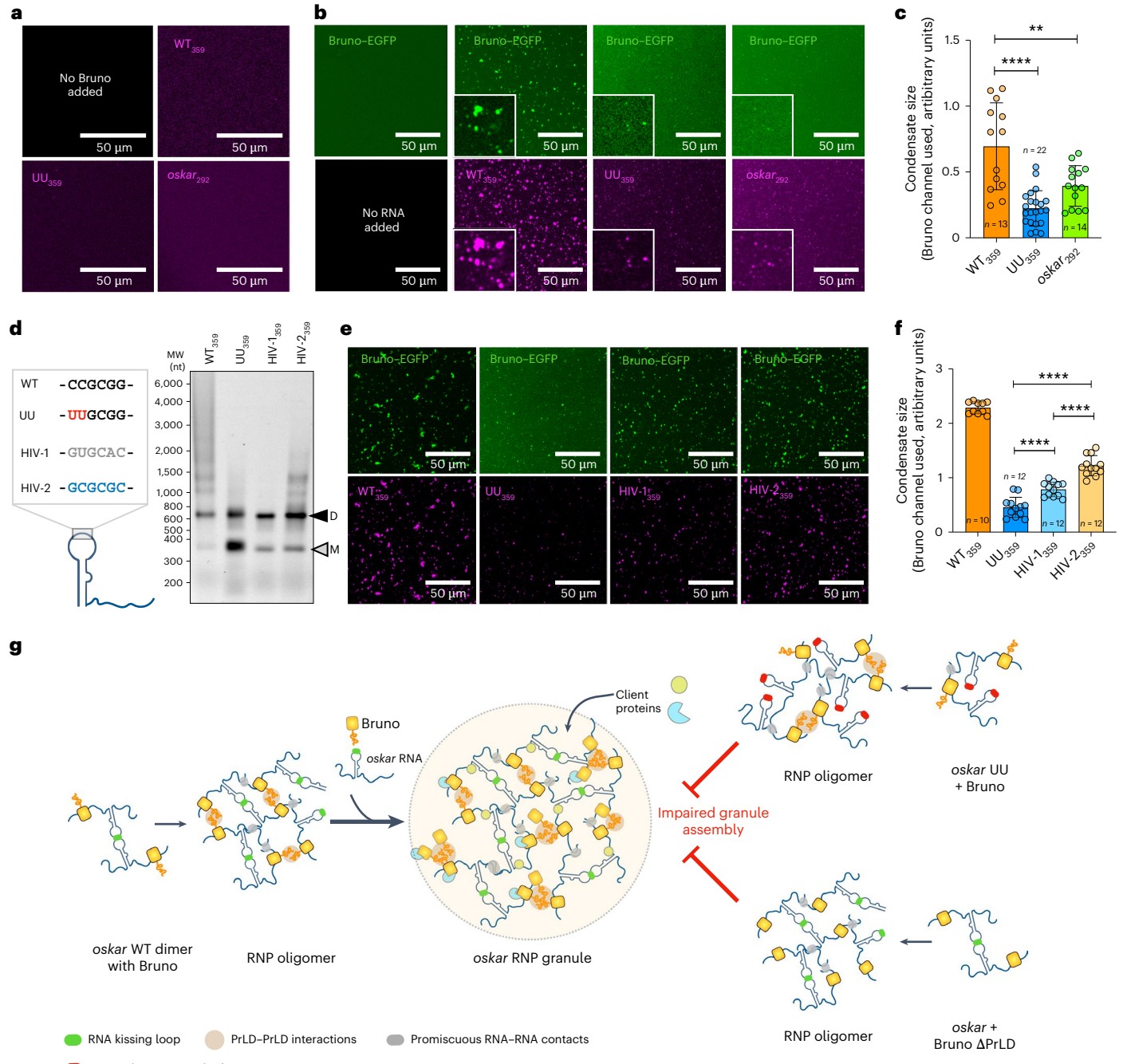

**Fig. 4 | The kissing-loop interaction is essential for condensate assembly with Bruno. a,b**, In vitro phase separation assay using 100 nM of the indicated atto633-labelled RNAs (magenta) alone (**a**) and 5 µM Bruno–EGFP (green) and 100 nM of the respective atto633-labelled RNAs (magenta) (**b**). **c**, Quantification of condensate size (based on the Bruno–EGFP channel) is plotted. Unpaired two-tailed Student's *t*-tests were used for comparisons. \*\**P* = 0.005, \*\*\*\**P* < 0.0001. The data in bar plot are presented as mean values ± standard deviation and *n* = 13, 22 and 14 FOVs comprising an average of 310, 107 and 142 particles from WT$_{359}$, UU$_{359}$ and *oskar*$_{292}$ RNAs, respectively, pooled from three independent replicates. The size of condensates is expressed as arbitrary (pixel) units. **d**, A schematic representation of the different kissing-loop sequences used. The dimerization and higher-order assembly are indicated by the atto633-labelled 359 nucleotide long RNAs detected by gel electrophoresis. 'D' indicates the dimeric form of the RNAs, and 'M' represents the monomer. MW, molecular weight marker.

**e**, In vitro condensate assembly using 5 µM Bruno–EGFP (green) and 100 nM of the respective atto633-labelled RNAs (magenta). **f**, Quantification of condensate size (based on the Bruno–EGFP channel). Unpaired two-tailed Student's *t*-tests were used for comparisons. \*\*\*\**P* < 0.0001. The data in the bar plot are presented as mean values ± standard deviation and *n* = 10, 12, 12 and 12 FOVs comprising an average of 300, 139, 200 and 205 particles from WT$_{359}$, UU$_{359}$, HIV-1$_{359}$ and HIV-2$_{359}$ RNAs, respectively, pooled from three independent replicates. The size of condensates is expressed as arbitrary (pixel) units. **g**, Schematic model showing the assembly of *oskar* granules as a result of cooperative multivalent interactions between the scaffold molecules *oskar* RNA and Bruno. The model depicts how the combinatorial action of RNA–RNA (RNA kissing loop and promiscuous inter-RNA contacts), RNA–protein (sequence-specific binding of Bruno to the *oskar* 3′ UTR) and protein–protein (Bruno PrLD–PrLD) interactions drive oligomerization and phase separation of *oskar* RNP granules.

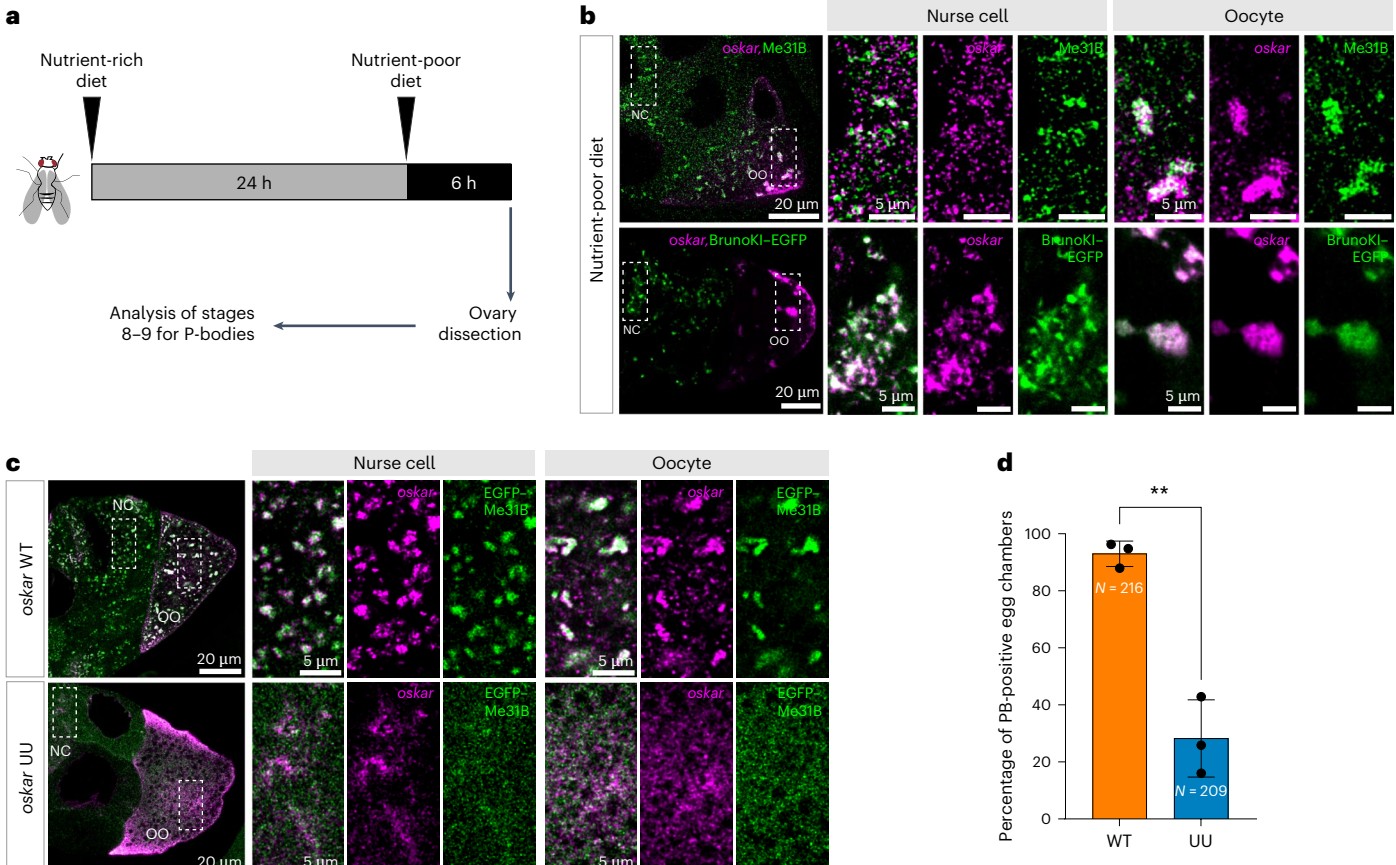

**Fig. 5 | Mutation in the *oskar* kissing loop impairs nutritional stress-induced P-body formation. a**, A schematic of the experimental procedure for nutrient deprivation of female flies. **b**, Co-detection of *oskar* mRNA (smFISH; magenta) and Me31B (immunofluorescence; green) in WT egg chambers (top) and *oskar* RNA smFISH (magenta) and Bruno–EGFP (green) in BrunoKI–EGFP egg chambers (bottom), after 6 h of nutrient deprivation. **c**, Confocal images of *oskar* RNA (magenta) and EGFP–Me31B (green) in mid-oogenesis stage egg chambers of *oskar* WT and *oskar* UU flies after 6 h of nutrient deprivation. Right: the boxed areas are enlarged. Note that, to allow better visualization of the fluorescent signal in the nurse cell compartment, brightness and contrast were adjusted during image processing. NC, Nurse Cell; OO, Oocyte. **d**, Bar plot showing the percentage of egg chambers forming P-bodies (PB-positive) in *oskar* WT and *oskar* UU flies. Three independent biological replicates were analysed, and egg chambers at stages 6–9 were scored. Each biological replicate is indicated with a black circle and represents a percentage of PB-positive egg chambers used for statistical comparison. *N* indicates a total number of egg chambers scored from all three biological replicates for *oskar* WT (*N* = 216) and UU (*N* = 209) flies. The data are presented as mean values ± standard deviation. \*\**P* = 0.0014. For the statistical analysis, an unpaired two-tailed *t*-test was used. The experiments were carried out in flies genetically null for endogenous *oskar* RNA.

kissing-loop interactions indeed contribute to condensate formation in our minimal system (Fig. 4e,f).

Multivalent interactions between protein and RNA are the driving force for RNP condensate network formation. Our experiments show that the intermolecular kissing-loop RNA–RNA interaction, as well as Bruno-driven oligomerization, co-scaffold *oskar* condensate assembly (Fig. 4g). Therefore, in the case of *oskar* granules, multivalency encoded by the PrLD–PrLD interactions acts synergistically with the intermolecular kissing-loop interaction to establish the *oskar* ribonucleoprotein network. Interfering with either Bruno phase separation, as we have previously shown, by deletion of the PrLD[14] or disrupting the RNA kissing-loop interaction (this study) is detrimental to *oskar* granule formation. These RNP assemblies provide the platform for recruitment of additional effector proteins that regulate *oskar*'s functions in germline development.

### Germline P-bodies depend on the kissing-loop interaction
Another class of RNP assemblies containing *oskar* mRNA is observed in the *Drosophila* female germline specifically upon nutrient deprivation[30,31] (Fig. 5a,b and Extended Data Fig. 7a,b). These stress-induced assemblies are referred to as processing bodies (P-bodies)[30], as they share protein components such as Me31B/Dhh1/

DDX6 with the yeast and mammalian P-bodies[32]. The localization of *oskar* mRNA to P-bodies is specific, as endogenous *bicoid* and *gurken* mRNAs were not enriched in these stress-induced assemblies (Extended data Fig. 7c). We observed that in addition to endogenous *oskar*, Bruno also localized to P-bodies (Fig. 5b and Extended data Fig. 7d). Furthermore, while the transgenic *oskar* WT RNA localized to P-bodies in absence of endogenous *oskar* RNA (Extended data Fig. 7e), more than 75% of egg chambers expressing *oskar* UU failed to form P-bodies after 6 h of nutrient deprivation (Fig. 5c,d). Our observations indicate that *oskar* mRNA is an integral and essential component of these assemblies. While a mechanistic understanding of how these large RNPs form under conditions of nutritional stress is lacking, the fact that a two-nucleotide substitution in the kissing loop of *oskar* disrupts P-body formation strongly suggests that, in addition to its architectural role in *oskar* transport granules, this high-affinity sequence-specific mRNA interaction mode also has a key role in P-body formation and highlights the importance of RNA–RNA interaction in driving diverse higher-order RNP assemblies in vivo.

## Discussion
Our findings demonstrate that a long mRNA molecule does not merely act as a passive scaffold that concentrates RBPs by sequence-specific

binding to promote multivalent protein–protein interactions that drive granule assembly. Rather, sequence-specific *trans* RNA–RNA interactions contribute to the granule network formation. The primary scaffold RBP, Bruno, binds to both WT and mutant RNA, but Bruno binding alone is insufficient to drive condensate assembly in absence of the kissing-loop interaction.

RNA kissing-loop interactions are well studied in retroviruses, whose genome is a dimer of two genomic RNAs (gRNAs) of positive polarity[33]. Dimerization of the gRNA is highly conserved and essential for the viral life cycle[29]. In the case of HIV-1, the dimerization initiation site comprises six nucleotide GC-rich palindrome which nucleates the intermolecular tertiary contact between two copies of the gRNA via a kissing-loop interaction (loose dimer), followed by a switch to an extended duplex (tight dimer) that stabilizes the dimer[28,34]. However, for *oskar* mRNA, a switch to an extended conformation does not occur in the oocyte, as evident from the predicted OES structure in vivo, inferred from mutational profiling and sequencing of dimethyl sulfate-modified RNAs (DMS-MapSeq)[35]. Additionally, super-resolution stochastic optical reconstruction microscopy imaging using fluorescence in situ hybridization probes spanning the entire length of *oskar* mRNA revealed a consistent drop in signal radius when probing near the stem loop compared with the rest of the RNA sequence, confirming sequence-specific *trans* RNA–RNA interactions being predominant in this region of the 3′ UTR[36].

Specific intermolecular RNA–RNA interactions can stem from canonical Watson–Crick base pairing between complementary RNA stretches or from Hoogsteen-type base pairing of G-rich tracts that form four-stranded G-quadruplexes, as reported for RNAs associated with repeat expansion disorders[10]. Specific RNA–RNA contacts also arise from kissing-loop interactions, which involve only very short palindromic sequences that can nevertheless impart exceptional mechanical stability to RNA dimers, as observed with artificial hairpins[37]. In fact, due to their cohesive nature, kissing loops are often engineered into RNAs to promote intermolecular contacts and facilitate three dimensional packing of RNAs for high resolution structural studies[38,39]. By modulating the strength of the kissing-loop base pairing using HIV sequences of varying GC content, we observed that the stronger specific RNA–RNA contacts were more effective in driving condensation. Interestingly, however, the HIV-2 sequence (GCGCGC), which is a scrambled version of the WT *oskar* sequence (CCGCGG), did not rescue higher-order RNA oligomerization and condensate formation to the same extent as WT *oskar*. This suggests that not only base composition but also nucleotide arrangement/geometry determines the strength of the kissing-loop interaction. Furthermore, while it remains a possibility that *oskar* RNA dimerization generates a new interaction platform for a specific client RBP that could further contribute to granule formation in vivo, our in vitro experiments emphasize the essentiality of the specific RNA interactions themselves in scaffolding the condensate network.

In addition to sequence-specific interactions, non-specific contacts are also highly prevalent in long RNA fragments, as seen in our in vitro experiments. Inside the cell, such promiscuous interactions presumably occur at high local concentrations of RNA, such as upon stress-induced release of mRNAs from polysomes[21], in transcription foci[40] or in RNA-rich condensates such as paraspeckles, which harbour up to 50 copies of the 23-kb-long *NEAT1_2* long non-coding RNA inside a ~300 nm condensate[41,42]. However, in our experiments, the promiscuous interactions not only failed to stabilize the dimeric species when exposed to thermal fluctuations but also could not drive condensate formation in vivo and in vitro upon the addition of the RBP. It is probable that within *oskar* granules, the prevalence of such non-specific RNA–RNA interactions depends on the availability of protein-free RNA segments. In contrast, the kissing-loop-mediated intermolecular interaction is crucial and acts as a specific RNA 'sticker'[22] and, together with Bruno PrLD–PrLD interactions, helps in physical crosslinking of the RNAs into an interconnected network with solid-like physical

properties[14], which is reinforced by additional promiscuous RNA–RNA contacts. Such a RNA–protein meshwork can facilitate partitioning of client proteins that can regulate condensate functions.

The possible advantage of sequence-specific RNA–RNA interactions in condensates is to facilitate compositional specificity with respect to the RNA species. This is critical for *oskar* granule function, as co-mixing with other maternal transcripts is detrimental for embryonic development[43,44]. Evidence of such RNA species-specific condensates comes from the filamentous fungus *Ashbya gossypii*, where co-assembling *SPA2* and *BNI1* mRNAs enrich in apically localized granules, whereas self-assembling *CLN3* mRNA forms distinct condensates around nuclei[45]. Interestingly, alteration of the secondary structure of *CLN3* leads to co-packaging of *CLN3* with the *SPA2/BNI1* granules, emphasizing the importance of RNA sequence- and structure-dependent interactions in specifying condensate composition. These condensates are either liquid-like or gel-like depending on the protein-to-RNA ratio, indicating that stable, sequence-specific RNA–RNA interactions do not determine the material state of these assemblies. In the *Drosophila* oocyte, a similar kissing-loop interaction leads to dimerization of *bicoid* mRNA, the maternal anterior determinant, which facilitates RNP formation with the double-stranded RNA binding protein (dsRBP) Staufen[46]. Our study adds evidence to the emerging role of specific and stable RNA–RNA interactions in scaffolding condensation. The fact that disrupting the kissing-loop interaction impaired the spatial regulation of *oskar* translation indicates that proper scaffolding by the mRNA is key to the establishment of translation repression in the transport granule condensates.

In summary, we demonstrate that a specific intermolecular RNA–RNA interaction in cooperation with an intrinsically disordered RBP can scaffold the assembly of a RNA–protein condensate that functions in germline specification and embryonic body axis patterning in *Drosophila*.

## Online content

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

## Methods

### Fly stocks and fly husbandry

The following *D. melanogaster* WT fly stocks were used: WT (*w^1118*) and *Oregon-R* (for the RNA affinity pulldown assay). To generate the *oskar* RNA null background the following fly stocks were used: *oskar^A87* (ref. [19]), *Df(3R)p^XT103* (ref. [13]) and *oskar^attP,3P3-GFP* (ref. [47]). The EGFP:Me31B fly line[48] was used to visualize P-bodies in the *Drosophila* germline. *oskar* WT and *oskar* UU transgenes were expressed in the *Drosophila* germline under control of the pUASP promoter using the *oskar*-GAL4 driver. The Bruno KI–EGFP line was a gift of Akira Nakamura[26].

All fly lines were maintained at 18 °C or 25 °C in vials or bottles on standard food (corn-based medium). The 3–6-day-old female flies, with typically half as many male flies, were transferred to vials with fresh yeast 24 h before ovary dissection. For nutrient deprivation, the flies were kept on standard food supplemented with dry yeast granules for 24 h at 25 °C (nutrient-rich conditions), transferred to fresh standard food without dry yeast (nutrient-poor conditions) and kept for 6 h at 25 °C. The ovaries were then dissected and processed for single-molecule fluorescence in situ hybridization (smFISH) and/or antibody staining.

### smFISH and immunostaining

Oligonucleotide probes (18–22 nt) spanning the coding sequence and the 3′ UTR of *oskar* mRNA were labelled using atto633 (Atto-Tec GmbH) as described previously[49]. The probe sequences are available in Supplementary Table 1. Freshly dissected *Drosophila* ovaries were fixed with 2% paraformaldehyde (PFA) in PBS with 0.05% Triton X-100 (Sigma) followed by two rounds of 10 min washes in PBS supplemented with 0.1% Triton X-100 (PBT). The ovaries were then pre-hybridized in hybridization buffer (HyB: 2× SSC, 1 mM EDTA buffer, 1 v/v% Triton X-100, 15 v/v% ethylene carbonate, 50 µg ml⁻¹ heparin and 100 µg ml⁻¹ single stranded DNA from salmon testes) for 10 min at 42 °C. Hybridization was performed at the same temperature in hybridization buffer (HyB) containing probe mix (2–3 nM per probe) for 2–3 h. After hybridization, the following 10 min × five-step washing protocol was performed: (1) HyB at 42 °C, (2) 1:1 HyB + PBT at 42 °C, (3) PBT at 42 °C, (4) pre-warmed PBT (42 °C) at room temperature and (5) PBT at room temperature. After washes, embedding or immunostaining was performed. To detect the EGFP-tagged proteins, native fluorescence of EGFP was visualized. To detect endogenous Me31B, immunostaining was performed following smFISH. The ovaries were incubated overnight with mouse α-Me31B antibody (1:200, Nakamura) in PBT supplemented with 1× β-casein at +4 °C, followed by three 10 min washes in PBT at room temperature. The ovaries were then incubated with secondary antibody (1:750; anti-mouse AlexaFluor633 IgG (H + L) highly cross adsorbed) in 1× β-casein/PBT for 2 h at room temperature, followed by three 10 min washes in PBT at room temperature. The ovaries were mounted in 25–50 µl 80% 2,2'-thiodiethanol (TDE) or in 85% glycerol supplemented with 2% propyl gallate. Counterstaining with 4,6-diamidino-2-phenylindole (DAPI) was done to visualize the nuclei.

### Immunofluorescence of ovaries and embryos

Freshly dissected ovaries were fixed in 4% PFA in PBS for 20 min, followed by extraction in permeabilization buffer (1% TritonX-100 in PBS) and blocking (0.5% bovine serum albumin (BSA), 0.3% TritonX-100 in PBS) for 1 h. For embryo collections, virgin female flies were mated with double the number of WT (*w^1118*) males for 2–3 days at 25 °C and maintained on apple juice agar plates and yeast paste. For Oskar immunofluorescence, 0–2-h-old eggs were collected, dechorionated using 50% bleach and washed extensively with distilled water, followed by fixation at the interface of 4% PFA and heptane for 20 min at room temperature. The embryos were devitellinized by vigorous shaking in a 1:1 mix of heptane and methanol. Fixed embryos were stored at −20 °C in 100% methanol. Before staining, the embryos were rehydrated in PBT, followed by blocking with western blocking reagent (Roche) in

PBT. The ovaries were blocked with 0.5% BSA in PBT. This was followed by incubation with rabbit α-Oskar (1:3,000) primary antibody overnight at 4 °C. Following three washes in PBT 15 min each, the samples were incubated with anti-rabbit Alexa Fluor 647 secondary antibody (1:750) for 2 h at room temperature in 10% goat serum in PBS. This was followed by three washes in PBT for 15 min each. The nuclei were stained with 4,6-diamidino-2-phenylindole (1:2,500 in wash buffer), and the samples were mounted in mounting media (80% glycerol, 2% propyl gallate).

### Microtubule staining of egg chambers

The ovaries were dissected in 1× Brinkley Renaturing Buffer 80 (80 mM PIPES, 1 mM EGTA, 1 mM MgCl₂) and fixed in 8% PFA in 1× Brinkley Renaturing Buffer 80 + 0.1% Triton X-100 for 20 min at room temperature. Post fixation, the ovaries were washed with 1× PBTB (1x PBS + 0.1% Triton X-100 + 0.2% BSA) five times, followed by staining with fluorescein isothiocyanate (FITC)-coupled mouse anti-α-tubulin antibody (SIGMA, F2168) at 1:200 dilution at 4 °C overnight. The samples were washed the following day in PBT and mounted on slides in a mounting medium (80% glycerol and 2% propyl gallate).

### Microscopy

All the images were acquired using a Leica SP8 TCS X confocal laser scanning microscope with a HC PL APO 63×/1.30 glycerol CORR CS2 glycerol-immersion objective. The images were deconvolved with the Huygens Essentials software or on the fly using the Leica Lightning module. For in vitro phase separation experiments, the images were acquired with an HC PL APO 40×/1.10 W CORR CS2 water-immersion objective without any further deconvolution.

### Embryonic cuticle preparations

A total of 15–20 virgin females of *oskar* WT or *oskar* UU were mated with double the number of *w^1118* males and fed with yeast paste for 2–3 days at 25 °C. Before egg collection, the flies were allowed to lay eggs overnight on apple juice agar plates in cages. The following day, the plates were collected, and the eggs aged for another 24 h at 25 °C. After collection, the eggs were dechorionated with 50% bleach for 2 min to remove the egg shell, washed extensively with distilled water and transferred to glass slides. The excess water was drained off, and the embryos were mounted in Hoyer's medium and lactic acid (Sigma), covered with a cover slip and baked overnight at 65 °C and imaged at 20× magnification using a bright-field microscope.

### In vitro transcription, fluorescent labelling and visualization

DNA templates for the in vitro transcription (IVT) reactions were prepared by polymerase chain reaction using T7-forward primer and gene specific reverse primers and extracted by gel purification. The primer sequences are available in Supplementary Table 2. IVT was performed with MegaShortScript T7 Kit (Invitrogen) for the 359- and 67-nucleotide-long fragments, and subsequent RNA purification was performed by acidic phenol–chloroform extraction. For the full length *oskar* 3′ UTR (WT and UU), MegaScript T7 Kit (Invitrogen) was used, and RNA was recovered by lithium chloride precipitation.

For fluorescent labelling, the IVT reaction was doped with 5-amino-allyl UTP (Biotium) at 1:10 (aminoallyl-UTP: UTP) followed by incubation of purified transcripts with threefold molar excess of atto633 NHS-ester (Atto-Tec GmbH) in 0.1 M NaHCO₃ at room temperature for 2 h, protected from light. The RNA was extracted using absolute ethanol and sodium acetate, pH 5.5 precipitation at −20 °C for at least 1 h and dissolved in ultrapure water (Invitrogen). The transcript size and integrity were assessed by gel electrophoresis of the RNA alongside RNA size markers (Riboruler HR and Riboruler LR, Thermo Scientific) and visualized using SYBR safe stain (473 nm) or by fluorescent imaging (635 nm) of atto633-labelled transcripts in a Typhoon biomolecular imager.

For visualization of RNA under indicated conditions, the labelled RNA was incubated at room temperature for 15 min in the indicated buffers and electrophoresed on a 0.8–1% agarose gel (0.5× Tris-borate-ethylenediaminetetraacetic acid buffer (TBE)) run at 100 V at 4 °C. For the smaller OES fragments, native 6% acrylamide gels (Novex TBE–urea gels, Invitrogen) were often used. The following buffer conditions were used: stringent zero salt buffer (20 mM Tris–HCl pH 7.5, 5% glycerol, 0.5 mM tris(2-carboxyethyl)phosphine (TCEP)), EMSA buffer (20 mM Tris–HCl pH 7.5, 150 mM NaCl, 2 mM MgCl₂, 5% glycerol, 0.5 mM TCEP) and high salt buffer (20 mM Tris–HCl pH 7.5, 300 mM NaCl, 5 mM MgCl₂, 5% glycerol, 0.5 mM TCEP).

### 3′ end labelling of RNA with biotin
The in vitro transcribed RNAs were 3′ biotinylated using pCp-biotin (Jena Biosciences). Briefly, 2 µM transcript was incubated with tenfold molar excess of pCp-biotin in T4 RNA ligase buffer, 10% dimethylsulfoxide, 1 mM ATP, 16% PEG-8000 and 1 unit per microlitre T4 RNA ligase enzyme at 16 °C overnight. The RNA was recovered by acidic phenol–chloroform extraction and reconstituted with ultrapure water.

### RNA affinity capture assay
Equimolar amounts of 3′-biotinylated RNA were added to ovary Lysis Buffer (5 mM Hepes–NaOH pH 7.5, 1 mM MgCl₂, 50 mM KCl, 25 mM sucrose, 0.5% NP-40, 50–75 U of ribolock (Thermo Fisher Scientific), 0.5% Triton X-100 (Sigma), 1 tablet of EDTA-free protease inhibitor cocktail (Merck), 10 mM dithiothreitol) and incubated with 25 µl of pre-washed paramagnetic streptavidin beads (Dynabeads MyOne Streptavidin C1 beads, Invitrogen) at 4 °C for 2 h.

Meanwhile, the ovary lysate was prepared. To obtain an optimal amount of ovary lysate for the pulldown assay, 30–40 g of *Oregon-R* flies was collected[23]. A total of 3–4 ml of collected ovaries were lysed in the lysis buffer using a Dounce homogenizer. The lysate was clarified and incubated with Avidin-agarose beads at 1/50th of the lysate volume for 30 min at 4 °C to remove endogenous biotinylated proteins. After removing the Avidin-agarose by mild centrifugation, the lysate volume was adjusted to 12 ml using the lysis buffer, and the lysate divided into six equal halves for the no RNA control, WT₃₅₉ and UU₃₅₉ conditions in duplicates. The respective bead-conjugated RNA was added to the lysate and incubated at 4 °C for 1 h in a nutator. Subsequently, 3 × 10 min washes were performed in the wash buffer (25 mM Hepes–NaOH pH 7.5, 1 mM MgCl₂, 150 mM KCl, 25 mM sucrose, 0.5% NP-40). In the final wash, 80% of the sample was eluted in Laemmli buffer supplemented with β-mercaptoethanol for western blotting, and the remaining 20% was used for RNA extraction using Trizol LS for quantiative polymerase chain reaction (qPCR)-based detection of pulled-down RNA levels.

### Western blotting
Western blotting was performed using the following primary antibodies: rabbit α-Bruno (1:1,000, in-house), rabbit α-Staufen (1:5,000, in-house), rabbit α-PTB (1:2,000, in-house) and rabbit α-Egalitarian (1:2,500)[50]. Anti-rabbit secondary antibody conjugated with horseradish peroxidase (GE Healthcare) was used for chemiluminescence-based detection.

### RNA extraction, complementary DNA synthesis and qPCR
SuperScript III First-Strand Synthesis System SuperMix (Invitrogen) was used for first-strand cDNA synthesis from isolated RNA using the manufacturer's instructions. Random hexamers were used for cDNA synthesis and the following gene specific primers for detecting *oskar* 359 fragment in the affinity-purified samples: 5′-GCGCGATTTTCGTCTTTCTGTTTC-3′ (forward) and 5′-GTAGCACAGTGTAGAATTCTGGCG-3′ (reverse).

### Protein purification
The pCoofy63-BrunoFL-EGFP (6xHis-SumoStar-BrunoFL-EGFP-TwinStrep) construct was used for expression and purification of Bruno as described previously[14]. Briefly, half a litre of Sf-21 insect cells were infected at a 0.5–0.7 × 10⁶ cell ml⁻¹density, with the recombinant baculovirus stock at a ratio of 1:100 and the cells collected 72 h post-infection. The cell pellet was flash frozen and stored at −80 °C. For purification, the pellet was resuspended in lysis buffer (20 mM Tris–HCl pH 7.5, 500 mM NaCl, 1 mM EDTA supplemented with 0.01% Triton X-100, 1× tablet of Complete Mini Protease Inhibitor cocktail (Roche), 2 mM MgCl₂) for 10 min on ice, followed by digestion with Benzonase (Sigma) for digestion of RNA/DNA and lysed using a microfluidizer. The lysate was clarified by centrifugation at 16,000*g* at 4 °C for 20 min, and the protein was affinity purified using a StrepTrap HP column by the C-terminal TwinStrep tag. The column was washed with five to six column volumes of wash buffer (20 mM Tris–HCl pH 7.5, 500 mM NaCl, 1 mM EDTA) and eluted in wash buffer supplemented with 2.5 mM desthiobiotin (Sigma). The protein-enriched fractions were pooled and dialysed overnight to remove EDTA and desthiobiotin. The following day, the protein was concentrated to 5 ml and subjected to size exclusion chromatography using a HiLoad 16/600 Superdex 200 pg column in storage buffer (20 mM Tris–HCl pH 7.5, 300 mM NaCl, 2 mM MgCl₂, 5% glycerol, 0.5 mM TCEP). The desired fractions were collected and concentrated using 50 kDa molecular weight cut-off (MWCO) concentrators (Amicon), aliquoted and flash frozen for storage at −80 °C. Importantly, during concentrating the protein post-size exclusion chromatography, the sample was frequently checked under a fluorescence microscope to make sure that no phase separation occurs.

### EMSAs
An EMSA was carried out as described previously[14,25]. A total of 50 nM of atto633-labelled *oskar* RNA construct was incubated with increasing concentrations of BrunoFL–EGFP for 20 min at room temperature in EMSA buffer (20 mM Tris–HCl pH 7.5, 150 mM NaCl, 2 mM MgCl₂, 5% glycerol, 0.5 mM TCEP) and resolved on a cold 0.8% agarose gel (0.5× TBE) run at 100 V at 4 °C. Imaging of the gel was performed in a Typhoon biomolecular imager.

### In vitro phase separation assays
All in vitro phase separation assays were carried out in assay buffer (20 mM Tris–HCl pH 7.5, 150 mM NaCl, 2 mM MgCl₂, 5% glycerol, 0.5 mM TCEP) and reactions spotted on 96-well non-binding µclear plates (Greiner Bio-One) for microscopy. A frozen aliquot of the protein was thawed and centrifuged at high speed to get rid of pre-formed aggregates, and the protein concentration was measured and the reaction assembled with or without 100 nM of atto633-labelled RNAs. The 6xHis-SumoStar tag was maintained during the experiments. The details of the individual experiments are indicated in the respective figure legends.

### Image analysis
Analysis and processing of acquired images were performed using Fiji. Maximum intensity projections of image stacks and histogram generation, as well as lane profiles, of the agarose gels and EMSAs were also calculated with Fiji.

**Quantification of fluorescence intensity.** To quantify *oskar* mRNA, two-dimensional sum projections were prepared for the oocyte and nurse cell compartments using on average 20 *z*-planes with a *z*-step size of 0.9 µm. For a region of interest, the raw integrated density of smFISH signal was measured, and the intensities were calculated as:

$$I = \text{raw integrated density}/(\text{area}(\mu m^2) \times z\text{-stack size }(\mu m)).$$ All representative images are two-dimensional maximum projections of, on average, two to three planes of acquired *xyz*-stack, unless indicated otherwise.

**Quantification of *oskar* localization.** *oskar* mRNA distribution was analysed using the CortAnalyis Fiji plugin as previously described elsewhere[14,47,51].

**Quantification of *oskar* partition coefficient in the oocyte.** Intensity-based segmentation of the granules was carried out on the basis of the *oskar* smFISH signal for the *oskar* WT and *oskar* UU genotypes. After trials with several segmentation algorithms, the 'triangles' algorithm could reliably distinguish granules from the diffuse RNA signal in the dilute phase in case of *oskar* UU. The partition coefficient was calculated as the ratio of the mean intensity inside granules to that of the oocyte cytoplasm[8].

**Quantification of condensate parameters.** For quantification of particle size, the Bruno–EGFP channel was filtered with a Gaussian filter of radius of one pixel, the particles were segmented using the 'triangles' algorithm and the areas were calculated using the particle analysis in Fiji.

### Statistics and reproducibility

For all quantifications, statistical analyses were performed and the data plotted using Prism 9. The *P* values were calculated using unpaired two-tailed Student's *t*-test. A *P* value of <0.05 was considered significant; in the figures, $*P < 0.05$, $**P < 0.01$, $***P < 0.001$ and $****P < 0.0001$. The precise *P* values are indicated in the figure legends. No statistical methods were used to pre-determine sample sizes, but the sample sizes were similar to those reported in previous publications. The data distribution was assumed to be normal but was not formally tested. No data were excluded from the analyses. The experiments were not randomized, and the investigators were not blinded to the conditions of the experiments during data collection, analysis and outcome assessment. The sample size and replicates are indicated in the respective figure legends. Imaging (smFISH and immunostaining) experiments were performed on ovaries from at least four to five female flies (each containing ovaries with multiple egg chambers) and repeated at least thrice at different times. RNA affinity capture, RNA gel electrophoresis, EMSAs and in vitro phase separation experiments were repeated in at least three independent replicates at different times. The total number of fields of view (FOVs) used for quantification is indicated in the respective figure legends.

#### Reporting summary

Further information on research design is available in the Nature Portfolio Reporting Summary linked to this article.

### Data availability

Source data are provided with this paper. All other data supporting the findings of this study are available from the corresponding authors on reasonable request.

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

### Acknowledgements

We thank I. Gaspar for sharing his observation that *oskar* UU and WT flies respond differently to nutrient deprivation and providing R-scripts for visualization of particle segmentation data. We are grateful to J. Nair for assistance with fly crosses, A. Nakamura for providing the EGFP–Bruno flies and S. Heber for the purified SNAP-Staufen protein. We acknowledge P. Jagtap, J. Hennig and F. Wippich for insightful discussions. M.B. was supported by DFG-FOR 2333 grant EP 37/4-1 from the Deutsche Forschungsgemeinschaft (Germany) to A.E. and by the EMBL. B.R. was supported by a studentship of the Darwin Trust of Edinburgh and the EMBL. J.M. and A.E. acknowledge funding from the EMBL.

### Author contributions

M.B., B.R., J.M. and A.E. designed the study, interpreted the results and wrote the paper. B.R. established the conditions for nutrient deprivation of flies and collected ovary samples for biochemical analysis. M.B. and B.R. performed the in vivo experiments and imaging analysis. M.B. carried out the in vitro experiments and analysed the data. M.B. carried out the experimental work for the revision.

### Funding

### Competing interests

The authors declare no competing interests.

### Additional information

**Extended data** is available for this paper at https://doi.org/10.1038/s41556-024-01519-3.

**Correspondence and requests for materials** should be addressed to Julia Mahamid or Anne Ephrussi.

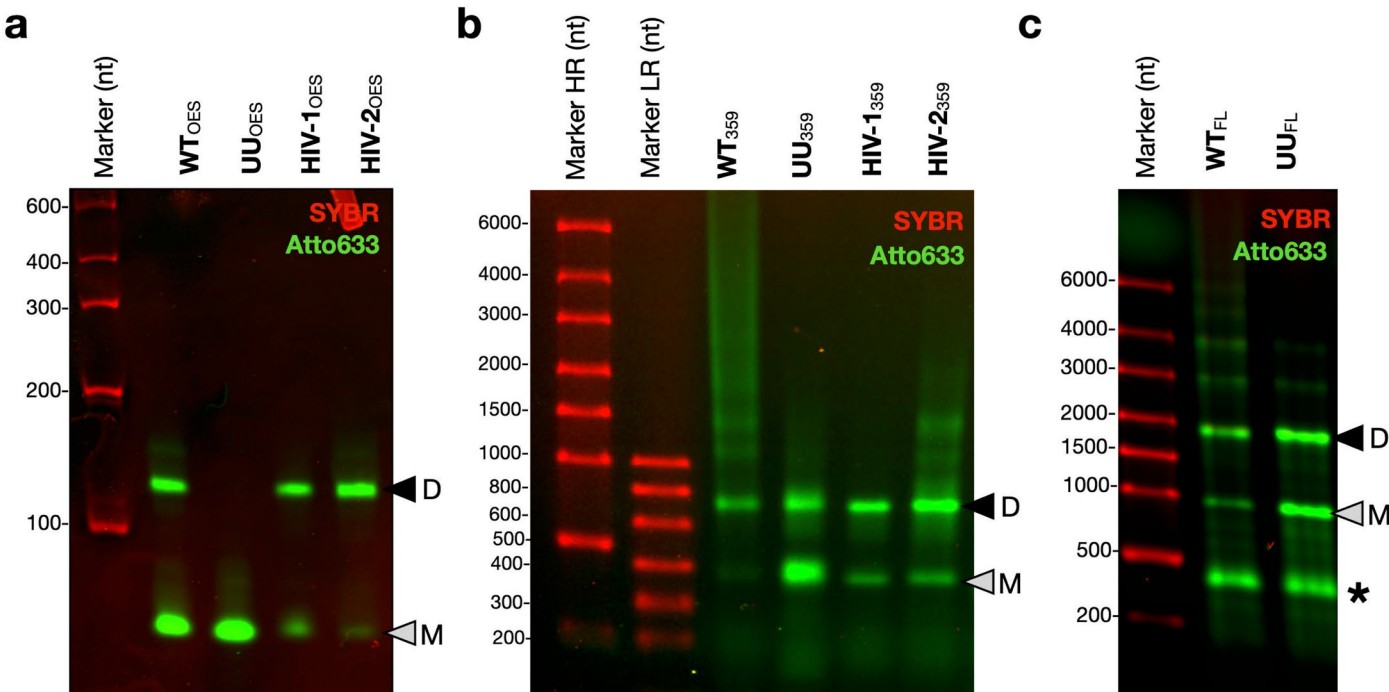

**Extended Data Fig. 1 | Characterization of *in vitro* transcribed RNAs used in this study.** RNA gel electrophoresis of all the *in vitro* transcribed RNAs used in the study, shown along with appropriate RNA markers (nt. = nucleotides). D indicates the dimeric form of the RNAs and M represents the monomer. **a**, the 67 nucleotide OES; **b**, the 359 nucleotide long RNA fragments. Note that the atto633 channel is also shown (greyscale) in Fig. 4d. **c**, the full length 3′UTR (1 kb) of *oskar* RNA. Asterisk (*) indicates an RNA species produced upon premature termination of the T7 *in vitro* transcription reaction only in the case of full length *oskar* 3′UTR. RNA sequencing identified that this sequence maps to bases 1–308 of the 3′UTR. As this fragment is upstream of the OES and is absent from the 359 fragment (**b**) used in the majority of the experiments, it does not affect the conclusions presented in this study.

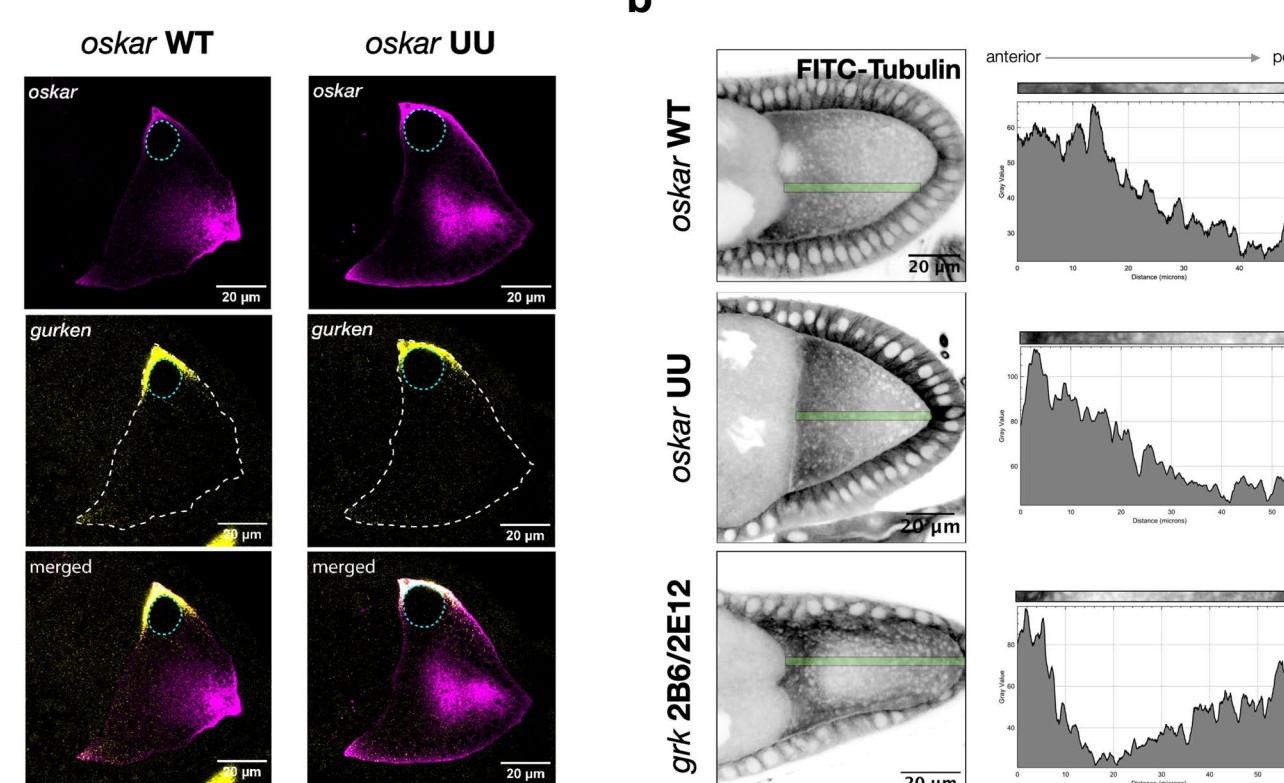

**Extended Data Fig. 2 | The kissing-loop mutation does not affect oocyte polarity. a**, smFISH images of *oskar* (magenta) and *gurken* (yellow) in *oskar* WT and UU egg chambers indicate that oocyte polarity is intact in both genotypes. Also note the correct anterior-lateral position of the nucleus (cyan dotted line), another marker of oocyte polarity. **b**, FITC-coupled anti-alpha-Tubulin antibody staining in the mid-oogenesis oocytes shows an anterior-to-posterior gradient of microtubules with a specific depletion at the posterior pole in both WT and

UU oocytes, further confirming that the UU mutation does not affect oocyte polarity. In contrast, in *gurken* mutant oocytes (*grk* 2B6/2E12), microtubules nucleate from all around the oocyte cortex and are depleted from the center of the oocyte, confirming that the polarised microtubule network is disrupted in these mutants[47]. Each intensity profile of FITC-Tubulin along the anteroposterior axis is derived from the boxed area (green) of the respective representative image of egg chambers of the indicated genotype.

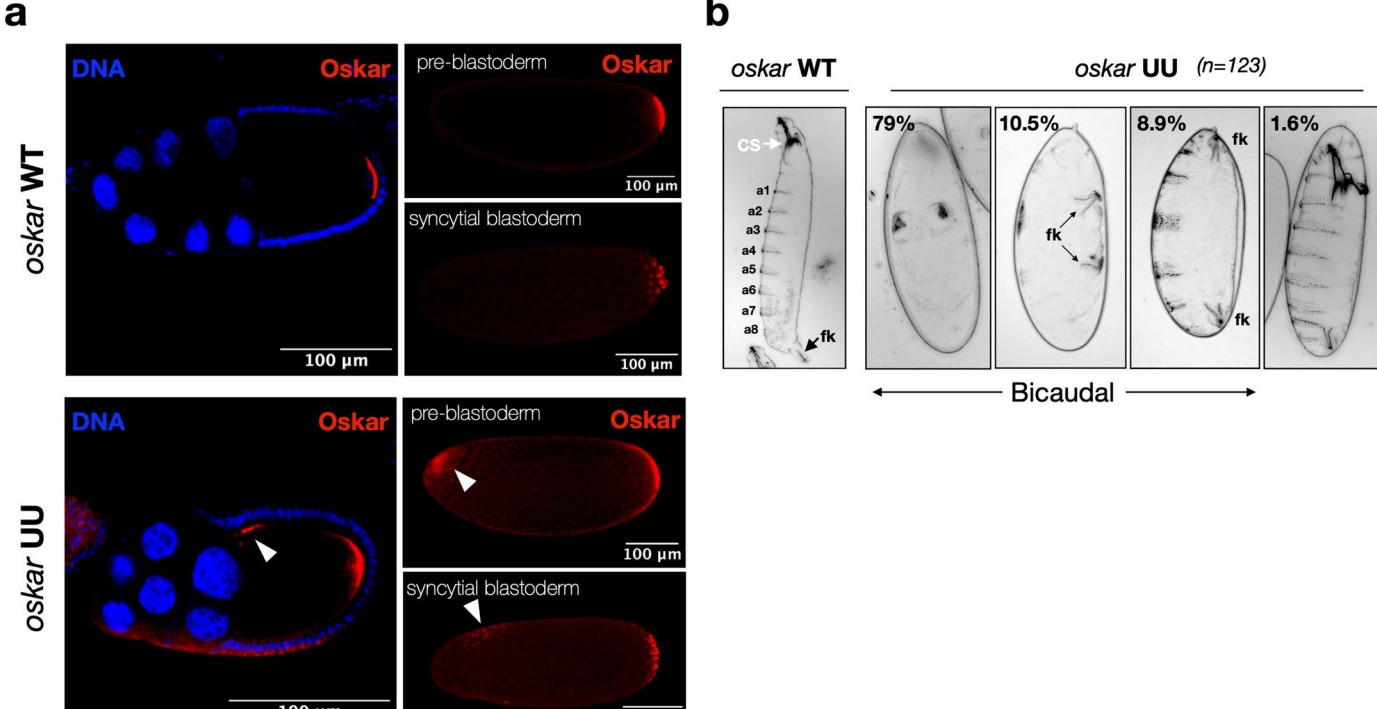

**Extended Data Fig. 3 | Effect of the kissing-loop mutation on *oskar* translation and embryonic development. a**, Immunofluorescence-based detection of Oskar protein in *oskar* WT and UU egg chambers (left) and embryos (right) shows ectopic accumulation of the protein (arrowheads) in the case of UU.

**b**, Representative cuticles showing strong bicaudal phenotype for the *oskar* UU mutants; anterior to the top, ventral to the left. a1–a8, abdominal segments; cs, head skeleton (white arrow); fk, filzkörper (black arrow).

**a**

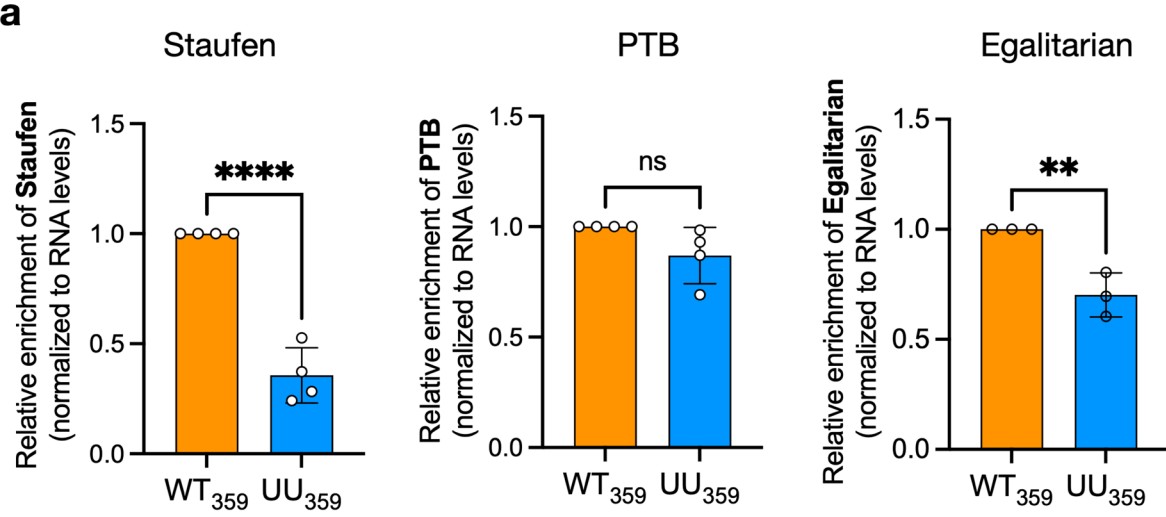

**b**

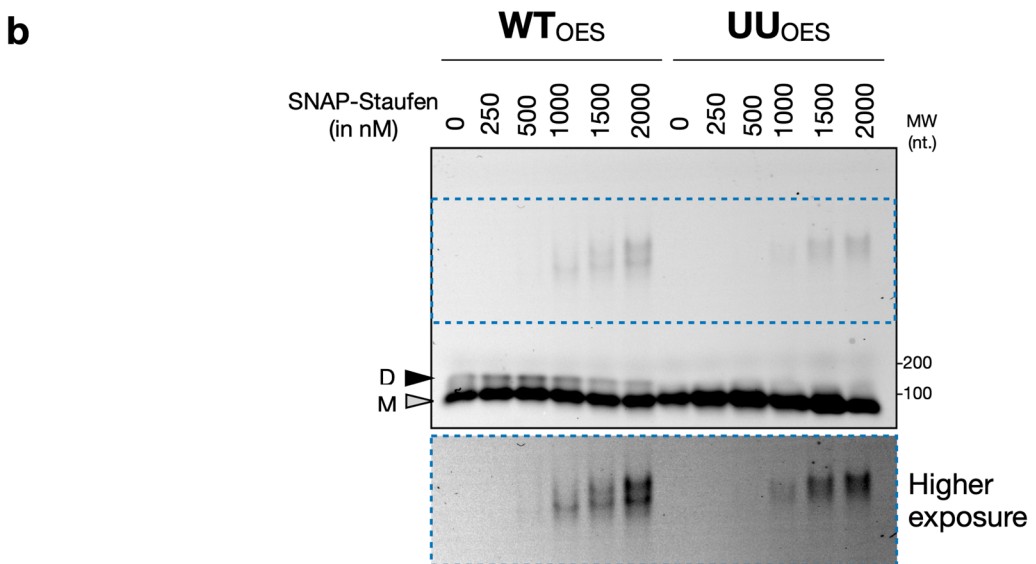

**Extended Data Fig. 4 | Effect of the kissing-loop mutation on association with** *oskar* **RNA-binding proteins. a**, Quantification of Staufen, PTB and Egalitarian levels associated with $WT_{359}$ or $UU_{359}$ RNAs based on western blots normalized to the recovered RNA levels from four, four and three independent replicates, respectively. Data are presented as mean values ± SD. Unpaired two-tailed Student's t-tests were used for comparisons. ns = non-significant, **$P = 0.0066$, ****$P < 0.0001$. **b**, EMSA of 50 nM atto633-labelled $WT_{67}$ or $UU_{67}$ RNA incubated with the indicated concentrations of SNAP-Staufen shows that both the WT and UU OES can bind Staufen *in vitro*. Lower image is a higher exposure of the upper half of the gel; cropped area is indicated by dotted blue box. nt. = nucleotides.

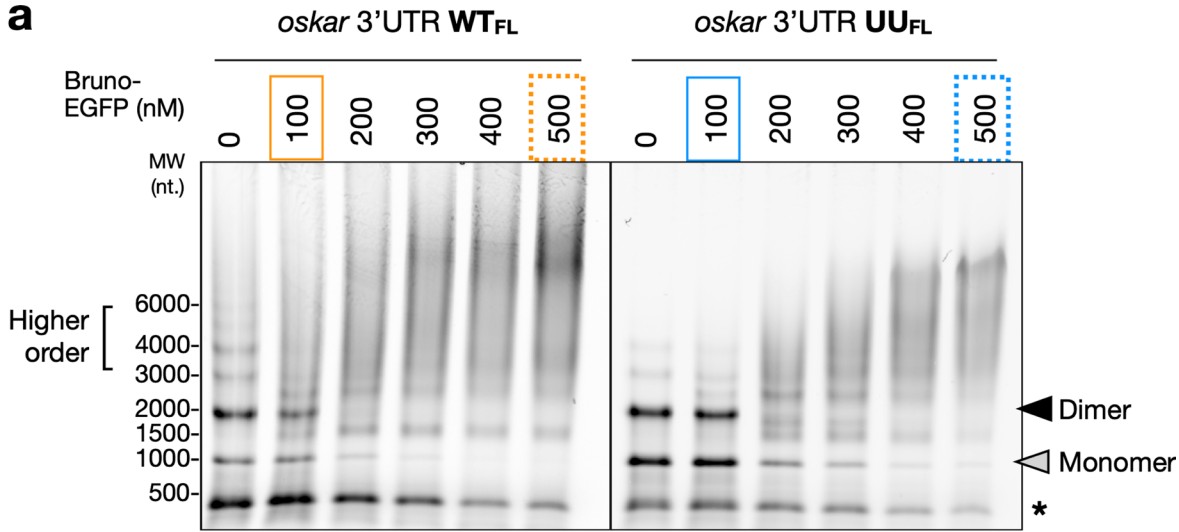

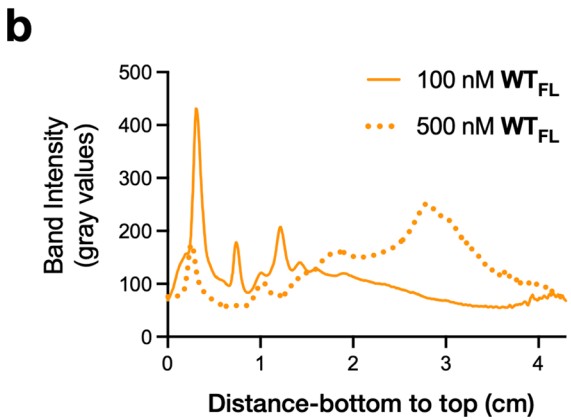
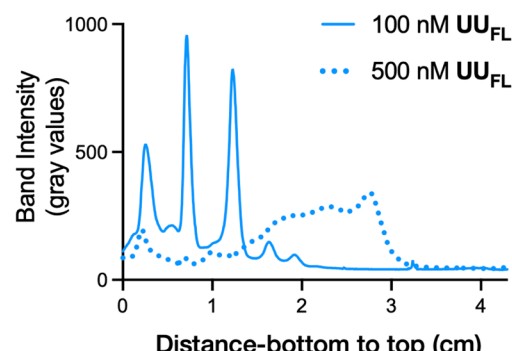

**Extended Data Fig. 5 | Higher-order complex formation of Bruno with wild-type and kissing-loop mutant *oskar* 3′UTR (full length). a**, EMSA of 50 nM atto633-labelled full length (FL) *oskar* 3′UTR incubated with the indicated concentrations of Bruno-EGFP. The WT and UU samples are from different regions of the same gel, as indicated by the black line. nt. = nucleotides.

**b**, Lane intensity profiles (from bottom to top of the gel) of the respective RNAs incubated with 100 or 500 nM of Bruno-EGFP. Owing to different degrees of labelling of the two RNAs, the band intensity values in the Y-axis are scaled. Note that the 308 bp fragment (*) also binds Bruno owing to the presence of BRE A and B sites. Representative results from three independent replicates are presented.

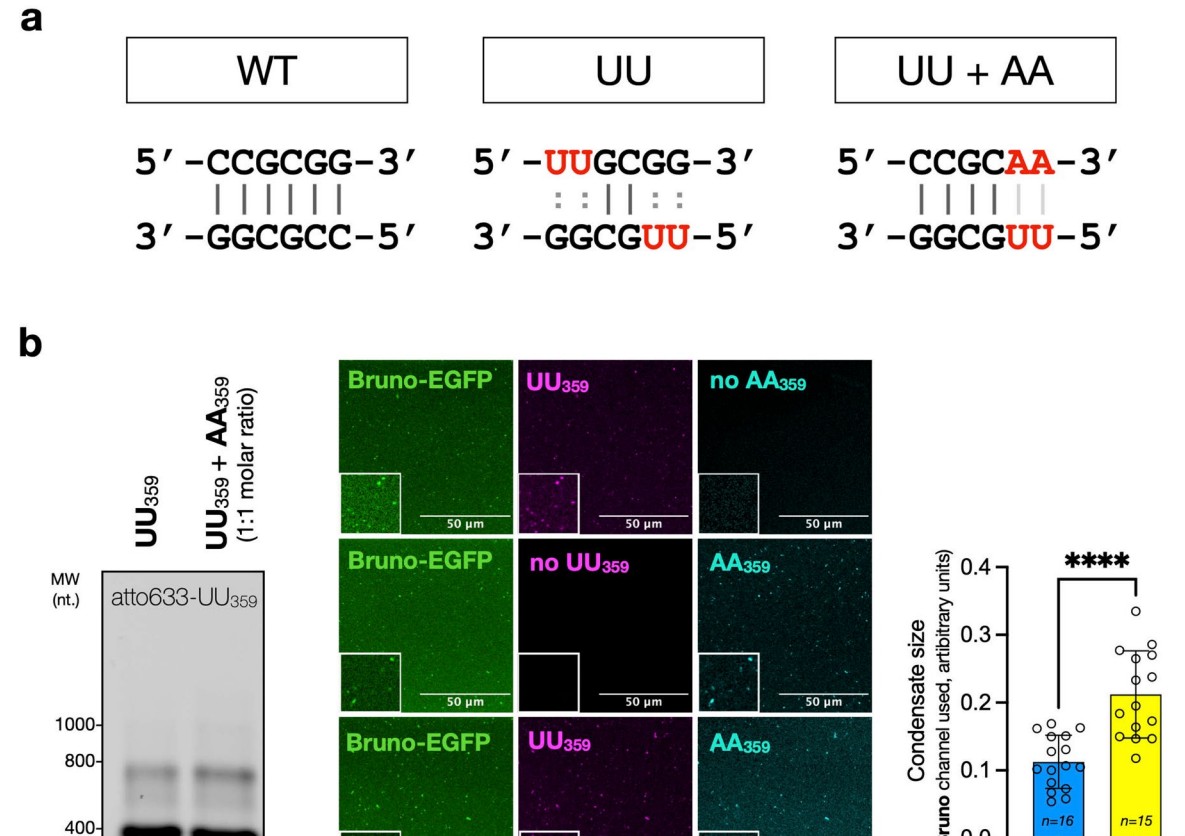

**Extended Data Fig. 6 | Restoring base pairing of the UU mutant only minimally rescues condensation with Bruno. a**, Schematic representation of the base pairing in the palindromic sequences forming the kissing-loop in the case of WT, UU and upon partial restoration of base pairing by introduction of the AA compensatory mutation. **b**, Dimerization is not appreciably rescued when atto633-labeled $UU_{359}$ and $AA_{359}$ RNAs are mixed at 1:1 molar ratio (fold rescue = 1.07 ± 0.047; n = 3 independent gel electrophoresis experiments). Addition of 5 µM Bruno-EGFP resulted in a small but significant increase in condensate formation as quantified in the right panel. **** $P < 0.0001$. Data are presented as mean values ± SD. Unpaired two-tailed Student's t-tests were used for comparisons. n denotes the number of FOVs; nt. = nucleotide bases.

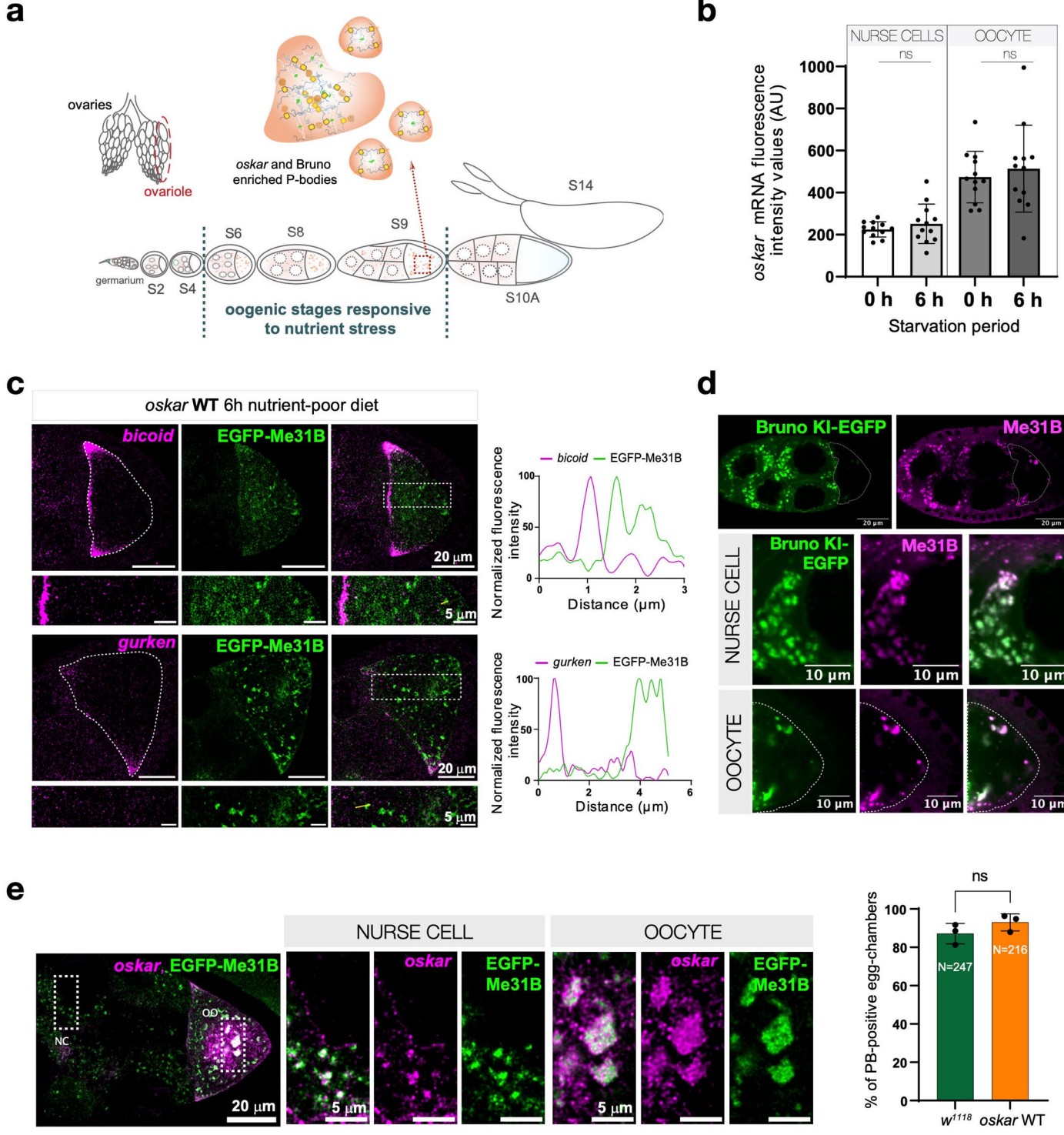

Extended Data Fig. 7 | See next page for caption.

**Extended Data Fig. 7 | Germline P-body formation is impaired in kissing-loop mutants. a**, Cartoon representation of a *Drosophila* ovariole showing the oogenesis stages analyzed for P-body formation in response to nutrient stress. **b**, Quantified intensities of *oskar* smFISH signal under nutrient-rich (0 h) and nutrient-poor (6 h) conditions in nurse cells and oocytes, indicating that *oskar* mRNA expression is not altered. Data are presented as mean values ± SD. ns = non-significant. For the statistical analysis ordinary one-way ANOVA applying Šídák's multiple comparison method was used. Each black dot indicates a single egg chamber analyzed. Fluorescence intensity values of nurse cells and oocyte compartment were obtained from 12 egg chambers maintained under the nutrient-rich condition and from 12 egg chambers subjected to 6 hours of nutrient deprivation condition. **c**, smFISH detection of *bicoid* mRNA (magenta; top panel) or *gurken* mRNA (magenta; bottom panel) in EGFP-Me31B (green) expressing *oskar* WT flies (in absence of endogenous *oskar* RNA). Boxed areas are enlarged below the images. Representative line profiles (yellow) are plotted on the right. **d**, Bruno localizes to P-bodies upon nutrient deprivation, as evident from the colocalization of Me31B immunofluorescence (magenta) and BrunoKI-EGFP (green) signal. **e**, Co-detection of *oskar* WT (magenta) and EGFP-Me31B (green) under nutrient-poor conditions. Boxed areas are enlarged on the right. The graph shows the percentage (%) of egg chambers forming P-bodies (PB-positive) in wild-type ($w^{1118}$) and transgenic *oskar* WT flies (in the absence of endogenous *oskar* mRNA). Bar plot displays mean of the quantified data with error bars showing the standard deviation (±SD). Three independent biological replicates were analyzed and egg chambers at stages 6–9 were scored. Each biological replicate is indicated with a black circle and represents percentage of PB-positive egg chambers used for statistical comparison. N denotes a total number of egg chambers scored from all three biological replicates for $w^{1118}$ (N = 247) and *oskar* WT (N = 216) flies. P-value = ns (non-significant, p = 0.2178) is determined by two-tailed unpaired two-tailed Student's t-test.

# Reporting Summary

## Statistics

For all statistical analyses, confirm that the following items are present in the figure legend, table legend, main text, or Methods section.

| n/a | Confirmed | |
|---|---|---|
| ☐ | ☒ | The exact sample size (*n*) for each experimental group/condition, given as a discrete number and unit of measurement |
| ☐ | ☒ | A statement on whether measurements were taken from distinct samples or whether the same sample was measured repeatedly |
| ☐ | ☒ | The statistical test(s) used AND whether they are one- or two-sided *Only common tests should be described solely by name; describe more complex techniques in the Methods section.* |
| ☒ | ☐ | A description of all covariates tested |
| ☒ | ☐ | A description of any assumptions or corrections, such as tests of normality and adjustment for multiple comparisons |
| ☐ | ☒ | A full description of the statistical parameters including central tendency (e.g. means) or other basic estimates (e.g. regression coefficient) AND variation (e.g. standard deviation) or associated estimates of uncertainty (e.g. confidence intervals) |
| ☐ | ☒ | For null hypothesis testing, the test statistic (e.g. *F*, *t*, *r*) with confidence intervals, effect sizes, degrees of freedom and *P* value noted *Give P values as exact values whenever suitable.* |
| ☒ | ☐ | For Bayesian analysis, information on the choice of priors and Markov chain Monte Carlo settings |
| ☒ | ☐ | For hierarchical and complex designs, identification of the appropriate level for tests and full reporting of outcomes |
| ☒ | ☐ | Estimates of effect sizes (e.g. Cohen's *d*, Pearson's *r*), indicating how they were calculated |

*Our web collection on statistics for biologists contains articles on many of the points above.*

## Software and code

Policy information about availability of computer code

| Data collection | Leica SP8 TCS X for image acquisition |
|---|---|
| Data analysis | Fiji 2.3.0/1.53q, CortAnalyis Fiji plugin, Graph Pad Prism v 9, Huygens Essentials software, Leica Lightning module of Leica SP8 TCS X |

For manuscripts utilizing custom algorithms or software that are central to the research but not yet described in published literature, software must be made available to editors and reviewers. We strongly encourage code deposition in a community repository (e.g. GitHub). See the Nature Portfolio guidelines for submitting code & software for further information.

## Data

Policy information about availability of data

All manuscripts must include a data availability statement. This statement should provide the following information, where applicable:
- Accession codes, unique identifiers, or web links for publicly available datasets
- A description of any restrictions on data availability
- For clinical datasets or third party data, please ensure that the statement adheres to our policy

Source data are provided with the study in the Supplementary Information file. All other data supporting the findings of this study are available from the corresponding authors on reasonable request.

## Research involving human participants, their data, or biological material

Policy information about studies with human participants or human data. See also policy information about sex, gender (identity/presentation), and sexual orientation and race, ethnicity and racism.

| | |
|---|---|
| Reporting on sex and gender | n/a |
| Reporting on race, ethnicity, or other socially relevant groupings | n/a |
| Population characteristics | n/a |
| Recruitment | n/a |
| Ethics oversight | n/a |

Note that full information on the approval of the study protocol must also be provided in the manuscript.

# Field-specific reporting

Please select the one below that is the best fit for your research. If you are not sure, read the appropriate sections before making your selection.

☒ Life sciences ☐ Behavioural & social sciences ☐ Ecological, evolutionary & environmental sciences

For a reference copy of the document with all sections, see nature.com/documents/nr-reporting-summary-flat.pdf

# Life sciences study design

All studies must disclose on these points even when the disclosure is negative.

| | |
|---|---|
| Sample size | For in vivo phenotypic analysis, three biological replicates were analyzed. For in vitro experiments, a minimum of three technical replicates were analyzed. For all experiments sample size was determined according to pilot experiments. Detailed description of sample size is provided in the figure legends. |
| Data exclusions | No data were excluded from the analysis. |
| Replication | A minimum of three replicates were performed for each experiment and used for analysis. Experiments provided reproducible data with similar results. Exact number of biological or technical replicates is indicated in the figure legends. |
| Randomization | Due to the nature of the experiments, no randomization was performed for the data assessment or analysis. |
| Blinding | Blinding was not performed due to the experimental condition set up. All the comparative samples were analyzed with the same parameters. |

# Reporting for specific materials, systems and methods

We require information from authors about some types of materials, experimental systems and methods used in many studies. Here, indicate whether each material, system or method listed is relevant to your study. If you are not sure if a list item applies to your research, read the appropriate section before selecting a response.

### Materials & experimental systems

| n/a | Involved in the study |
|---|---|
| ☐ | ☒ Antibodies |
| ☒ | ☐ Eukaryotic cell lines |
| ☒ | ☐ Palaeontology and archaeology |
| ☐ | ☒ Animals and other organisms |
| ☒ | ☐ Clinical data |
| ☒ | ☐ Dual use research of concern |
| ☒ | ☐ Plants |

### Methods

| n/a | Involved in the study |
|---|---|
| ☒ | ☐ ChIP-seq |
| ☒ | ☐ Flow cytometry |
| ☒ | ☐ MRI-based neuroimaging |

## Antibodies

| | |
|---|---|
| Antibodies used | Rabbit anti-Bruno, rabbit anti-Staufen, rabbit anti-PTB, rabbit anti-Egalitarian, mouse anti-Me31B |

| Validation | Rabbit anti-Bruno: Validated by Western blot of UV-crosslinking and RNA affinity pull down experiments (reference: Besse et al., Drosophila PTB promotes formation of high-order RNP particles and represses oskar translation, Genes & Development 23:195–207 (2009))

Rabbit anti-Staufen: Validated by immunofluorescence using egg chambers from wild type and staufen alleles (reference: St Johnston, et al., Staufen, a gene required to localize maternal RNAs in Drosophila eggs. Cell 66, 51-63 (1991))

Rabbit anti-PTB:  Validated by Western blot in PTB GFP-trap lines and PTB germline clones (reference: Besse et al., Drosophila PTB promotes formation of high-order RNP particles and represses oskar translation, Genes & Development 23:195–207 (2009))

Rabbit anti-Egalitarian: Validated by Western blot in Egl mutant alleles (reference: Mach, J. M. & Lehmann, R. An Egalitarian-BicaudalD complex is essential for oocyte specification and axis determination in Drosophila. Genes & development 11, 423-435 (1997))

Mouse anti-Me31B: Validated by immunofluorescence in Me31B null tissue sample (reference: Nakamura, A., Amikura, R., Hanyu, K. & Kobayashi, S. Me31B silences translation of oocyte-localizing RNAs through the formation of cytoplasmic RNP complex during Drosophila oogenesis. Development 128, 3233–3242 (2001)) |
|---|---|

# Animals and other research organisms

Policy information about studies involving animals; ARRIVE guidelines recommended for reporting animal research, and Sex and Gender in Research

| Laboratory animals | Drosophila melanogaster:
OregonR, w1118 was used as wild type. OregonR (Bloomington Stock #5) was used for mass extraction of ovaries for RNA affinity capture.

oskarA87 (Jenny, A. et al. A translation-independent role of oskar RNA in early Drosophila oogenesis. Development 133, 2827-2833, doi:10.1242/dev.02456 (2006)).

Df(3R)pXT103 (Lehmann, R. & Nusslein-Volhard, C. Abdominal segmentation, pole cell formation, and embryonic polarity require the localized activity of oskar, a maternal gene in Drosophila. Cell 47, 141-152, doi:10.1016/0092- 8674(86)90375-2 (1986)).

oskarattP,3P3-GFP (Gaspar, I. et al., An RNA-binding atypical tropomyosin recruits kinesin-1 dynamically to oskar mRNPs. EMBO J 36, 319-333, doi:10.15252/embj.201696038 (2017)).

EGFP:Me31B (Nakamura, A. et al., Me31B silences translation of oocyte-localizing RNAs through the formation of cytoplasmic RNP complex during Drosophila oogenesis. Development, 128, 3233–3242 (2001)).

oskar WT (Hachet, O. & Ephrussi, A. Splicing of oskar RNA in the nucleus is coupled to its cytoplasmic localization. Nature 428, 959-963, (2004)).

oskar UU (Jambor, H. et al., Dimerization of oskar 3' UTRs promotes hitchhiking for RNA localization in the Drosophila oocyte. RNA 17, 2049-2057, (2011)).

oskarGAL4 (Bloomington Stock #44242).

BrunoKI-EGFP (DGRC # 118625). |
|---|---|
| Wild animals | No wild animals were used in this study. |
| Reporting on sex | Reporting on sex was not applicable in this study. |
| Field-collected samples | No samples were collected from the field. |
| Ethics oversight | No ethical approval was needed for this study. |

Note that full information on the approval of the study protocol must also be provided in the manuscript.

