## [Peer Review File · Nature Cell Biology]

Peer Review Information

Journal: Nature Cell Biology

Manuscript Title: An architectural role of specific RNA-RNA interactions in oskar granules

Corresponding author name(s): Dr Anne Ephrussi

Editorial Notes:

Redactions – unpublished data	Parts of this Peer Review File have been redacted as indicated to maintain the confidentiality of unpublished data.
Redactions – confidential patient information	Parts of this Peer Review File have been redacted as indicated to maintain patient confidentiality.
Redactions – published data	Parts of this Peer Review File have been redacted as indicated to remove third-party material.
Redactions – reviewer opt-out	Parts of this Peer Review File have been redacted as indicated as we could not obtain permission to publish the reports of reviewer no. XX .
Reviewer comments in marked-up manuscript	In their review of the [first/second/third/...] version of this manuscript, reviewer no. XX added their comments to the manuscript file. These comments, excluding minor textual revisions, have been copied into this Peer Review File.

Reviewer Comments & Decisions:

Decision Letter, initial version:
--

*Please delete the link to your author homepage if you wish to forward this email to co-authors.

Dear Dr Ephrussi,

Your manuscript, "An architectural role of oskar mRNA in granule assembly", has now been seen by 3 referees, who are experts in Drosophila oogenesis (referee 1); RNAs and development (referee 2); and

biomolecular condensation (referee 3). As you will see from their comments (attached below) they find this work of potential interest, but have raised substantial concerns, which in our view would need to be addressed with considerable revisions before we can consider publication in Nature Cell Biology.

Nature Cell Biology editors discuss the referee reports in detail within the editorial team, including the chief editor, to identify key referee points that should be addressed with priority, and requests that are overruled as being beyond the scope of the current study. To guide the scope of the revisions, I have listed these points below. We are committed to providing a fair and constructive peer-review process, so please feel free to contact me if you would like to discuss any of the referee comments further.

In particular, it would be essential to:

- A) Test for whether the RNA stem loop interactions are sufficient for the proposed effects on condensation, or whether or not they may indirectly recruit potentially other proteins (Reviewers #1, #2).
- B) Assess effects on translation (Reviewer #3)
- C) Characterize effects of, and localization of, Bruno/Stau/Egl (Reviewers #1 and #2).
- D) All other referee concerns pertaining to strengthening existing data, providing controls, methodological details, clarifications and textual changes, should also be addressed.
- E) Finally please pay close attention to our guidelines on statistical and methodological reporting (listed below) as failure to do so may delay the reconsideration of the revised manuscript. In particular please provide:
 - a Supplementary Figure including unprocessed images of all gels/blots in the form of a multi-page pdf file. Please ensure that blots/gels are labeled and the sections presented in the figures are clearly indicated.
 - a Supplementary Table including all numerical source data in Excel format, with data for different figures provided as different sheets within a single Excel file. The file should include source data giving rise to graphical representations and statistical descriptions in the paper and for all instances where the figures present representative experiments of multiple independent repeats, the source data of all repeats should be provided.

We would be happy to consider a revised manuscript that would satisfactorily address these points, unless a similar paper is published elsewhere, or is accepted for publication in Nature Cell Biology in the

meantime.

- ensure that it conforms to our format instructions and publication policies (see below and <https://www.nature.com/nature/for-authors>).
- provide a point-by-point rebuttal to the full referee reports verbatim, as provided at the end of this letter.
- provide the completed Reporting Summary (found here <https://www.nature.com/documents/nr-reporting-summary.pdf>). This is essential for reconsideration of the manuscript will be available to editors and referees in the event of peer review. For more information see <http://www.nature.com/authors/policies/availability.html> or contact me.

When submitting the revised version of your manuscript, please pay close attention to our [href="https://www.nature.com/nature-portfolio/editorial-policies/image-integrity">Digital Image Integrity Guidelines](https://www.nature.com/nature-portfolio/editorial-policies/image-integrity). and to the following points below:

Nature Cell Biology is committed to improving transparency in authorship. As part of our efforts in this direction, we are now requesting that all authors identified as 'corresponding author' on published papers create and link their Open Researcher and Contributor Identifier (ORCID) with their account on the Manuscript Tracking System (MTS), prior to acceptance. ORCID helps the scientific community achieve unambiguous attribution of all scholarly contributions. You can create and link your ORCID from the home page of the MTS by clicking on 'Modify my Springer Nature account'. For more information please visit www.springernature.com/orcid.

This journal strongly supports public availability of data. Please place the data used in your paper into a

public data repository, or alternatively, present the data as Supplementary Information. If data can only be shared on request, please explain why in your Data Availability Statement, and also in the correspondence with your editor. Please note that for some data types, deposition in a public repository is mandatory - more information on our data deposition policies and available repositories appears below.

<https://mts-ncb.nature.com/cgi-bin/main.plex?el=A5C3CMG6A7BkfR3J4A9ftdUSptwIOtcY2Faor6M0M4gZ>

We would like to receive a revised submission within six months.

We hope that you will find our referees' comments, and editorial guidance helpful. Please do not hesitate to contact me if there is anything you would like to discuss.

Best wishes,

Daryl Jason David

Daryl Jason Verzosa David, PhD

Senior Editor, Nature Cell Biology
Nature Portfolio

Heidelberger Platz 3, 14197 Berlin, Germany
Email: daryl.david@nature.com
ORCID: <https://orcid.org/0000-0002-9253-4805>

Reviewers' Comments:

Reviewer #1:

Remarks to the Author:

It has been known that relatively weak, multivalent, and often promiscuous interactions between RNA-RNA, RNA-RBP, and IDR/PrID in RBPs promote the formation of biomolecular condensates. In this manuscript, Bose et al. report that a stable base-pairing interaction between the kissing loops of the SL2b in the *osk* RNA contributes to mesoscale granule formation in the *Drosophila* oocyte. The disruption of the kissing-loop interaction showed defects in *osk* RNA localization to the posterior pole of the oocyte. This *osk*-UU mutant RNA did not form mesoscale granules. The authors also showed that the 359-base *osk* RNA fragment containing the SL2b and Bruno (Bru) binding sequence pulled down Bru from ovary lysates. Notably, the strength of the interaction was dependent on RNA dimerization, which was also important for Bru-mediated condensation in both *in vivo* and *in vitro*. Furthermore, the authors showed that the kissing-loop interaction of *osk* RNA was crucial for the stress-induced formation of P-bodies that contain *osk* RNA and Bru. The authors propose that the sequence-specific intramolecular RNA-RNA interaction acts with IDR-containing RBPs in driving condensation.

A clear demonstration of the contribution of the specific base-pairing interaction between RNA molecules in phase separation is, to my knowledge, novel and should be a significant finding. A broad audience in developmental and molecular cell biology including many readers of *Nature Cell Biology* will be interested in this discovery, which will stimulate related research fields. Thus, the manuscript should be worthy of reporting in premiere-class journals.

The manuscript is well-written and easy to follow the authors' logical flow. Qualities of figures are generally high. I think that statistical processing is adequately operated.

However, it remains unclear from the current resolution of data whether the kissing-loop interaction of *osk* mRNA directly stimulates condensation, or the dimerization generates a new platform for a specific client RBP, which in turn contributes to condensation. Although the novelty of their finding will be unchanged in either case, I think that the authors need to discriminate between these two possibilities and describe a clearer picture of how a stable base-pairing interaction contributes to condensation. I would like to suggest several possible experiments below. As this field of research has been extremely complicated, there may be misunderstandings in my comments below. If the authors find such statements, please rebuttal to my suggestions.

Major concerns:

1. The authors showed that the *osk*-UU mutant did not dimerize, failed to form larger granules in the oocyte, and showed defects in its posterior localization. These results indicate the necessity of the kissing-loop interaction for proper *osk* RNA behaviors. I wonder if the dimerization through the kissing

loops in the SL2b is sufficient for *osk* RNA behaviors. That means, the authors could provide data on the restoration of the kissing-loop interaction can rescue the defects. The authors' group previously reported that the AA mutation in the loop acted as a compensatory mutant and restored the dimerization with the UU-loop (Jambor et al. RNA 2011). The authors can examine whether co-expression of the AA form of the transgene can rescue defects observed in the *osk*-UU mutant. Also, the authors could conduct whether the presence of both *osk*-UU359 and *osk*-AA359 RNAs promotes Bru-mediated condensation in vitro.

2. If the dimerization of *osk* RNA through the kissing-loop interaction is sufficient to promote condensation independently from the face-to-face duplication of the SL2b stem, the addition of an engineered kissing-loop sequence (lines 320-322) in the *osk*-UU RNA might rescue defects. I think that the authors could conduct these types of experiments.

3. In an in vitro assay, conditions of the reaction (components, concentrations, salts, temperature, etc.) can be optimized, and do not reflect in vivo situations. For example, in the reaction shown in Fig 3e, the addition of an RBP that interacts with SL2b might strongly stimulate condensation. Notably, the authors showed that the association of Stauf (Stau) to the *osk*359 RNA fragment was reduced in the UU mutant; the reduction level was similar to that observed for Bruno (Extended Data Fig. 2). Since Stau contains multiple dsRNA-binding domains (Ramos et al. EMBO J. 2000), it could interact with the stem structure in the SL2b (as has been predicted in Mohr et al. PLOS Genet. 2021) and the interaction might be enhanced when SL2b stems become tandemly oriented by the kissing-loop interaction.

The authors showed that the *osk*-UU RNA failed to localize to the posterior pole of the oocyte (Fig 1c). Given that Stau is crucial for the posterior localization of *osk* RNA in the oocyte, the defects might be caused by the reduced interaction between *osk* RNA and Stau (the idea is also supported by data shown in Extended Data Fig. 2). A large fraction of the *osk*-UU RNA also remained at the anterior and lateral cortex in the oocyte (Fig 1c), which suggests that the *osk*-UU RNP still associates with the active form of dynein-mediated transport machinery even in the oocyte. It has been reported that a dynein machinery component Egalitarian (Egl) antagonizes with Stau for localization of *osk* RNA within the oocyte (Mohr et al. PLOS Genet. 2021). The authors' group also recently reported an antagonized relationship between Stau and Egl (Gáspár et al. J. Cell Biol. 2023). Furthermore, the authors' group has reported that Egl is a candidate RBP that interacts with the SL2b stem for dynein-mediated transport (Jambor et al. RNA 2014). These observations additionally support the idea that the defects observed in the *osk*-UU mutant are caused by the reduction of the *osk* RNA-Stau interaction. In this scenario, the kissing-loop interaction strengthens the interaction between the tandemly oriented SL2b stems and Stau, leading to the displacement of Egl from each stem. I think that the authors could examine and discuss the potential roles of Stau and Egl in SL2b-mediated condensation and posterior localization of *osk* RNA. For example, the authors could test whether Stau interacts with the SL2b stems, especially with the face-to-face duplicated form. The authors could also examine whether the association of Egl to the *osk*359 RNA in

the pull-down assay (Fig 2d) is stronger in the *osk*-UU mutant RNA (due to lack of Stau-mediated interference).

Minor points:

1. The authors show that the *osk*-UU RNA does not form large granules (Fig 1e). I wonder if the wild-type *osk* RNA forms granules in *stau* mutant oocytes.

2. Lines 107-110: The authors argue that the polarity in the oocyte is intact in the *osk*-UU mutant by examining *grk* RNA localization to the anterior-dorsal corner of the oocyte (Extended Data Fig. 1). Considering the microtubule organization and *osk* RNP behaviors within the oocyte (Zimyanin et al Cell 2008; Parton et al. J. Cell Biol. 2011), the data on *grk* RNA localization alone is insufficient to judge the polarity in the oocyte. I think that the authors will have to examine the posterior enrichment of Kinesin (either by endogenous protein detection or using the kinesin motor domain- β -galactosidase marker) or directly examine microtubule organization (e.g. EB1-GFP imaging).

3. Extended Data Fig. 2: I suggest that the authors should conduct the experiment more than triplicates and conduct statistical analysis.

4. The authors previously reported that the *osk* RNP is a solid-like condensate (Bose et al. Cell 2022). Does a relatively strong and stable base-pairing interaction between the kissing loops of the *osk* RNA contribute to its solid-like status? I think that many researchers tackling the condensation issues feel that stable RNA-RNA interactions provide a negative effect driving condensation. If possible, I think it would be wonderful if the authors could discuss this issue.

Reviewer #2:

Remarks to the Author:

In this manuscript, the authors explore the role of a palindromic sequence, called the kissing-loop, located within the *oskar* mRNA 3'UTR in mediating RNA-protein granule assembly. Through a combination of *in vitro* RNA biochemistry and *in vivo* cell biology, they discover that the kissing-loop helps to promote dimerization of *oskar* mRNA and ultimately drive the formation of higher order ribonucleoprotein assemblies containing the RNA binding protein Bruno. Additionally, they show that the *oskar* mRNA, and specifically the kissing-loop, is required for the formation of P bodies upon nutrient deprivation. The study provides important and timely insight into the role of specific intermolecular RNA-RNA interactions in bimolecular condensate formation and as such will likely appeal to a broad audience. The experiments are sound and the conclusions are largely supported by the data (see comments below).

Specific Comments

Is it possible that protein interactions with the kissing-loop underly its role in granule assembly, rather than RNA dimerization? Presumably not Bruno based on your results but possibly other RBPs, such as Me31B. One way to test this would be to alter the nucleotide sequence of the kissing-loop domain in a way that doesn't substantially impact RNA dimerization.

Fig. 1b, 2a, 2b, 3b, 3c. Size markers in the gel images would be useful.

Fig. 2d. Why is there no variation in wt in the bar plot? Presumably this is related to normalization but the variation should nonetheless be displayed (standard deviation from the mean) and of course the statistical analysis should also take into account variation in wt.

Fig. 5b and associated results. In concluding that Bruno localizes to P bodies, you don't show co-localization of Bruno with typical P body markers. Presumably the logic here is that because oskar overlaps with Me13B and Bruno overlaps with oskar, that Bruno is also in P bodies?

Reviewer #3:

Remarks to the Author:

In the paper entitled, "An architectural role of oskar mRNA in granule assembly" Bose et al propose that RNA oligomerization dictated by the kissing stem loops of oskar mRNA is responsible for 2 varieties of condensate formation and oskar RNA localization. To my knowledge this represents the first conclusive role for RNA oligomerization in promoting condensation, not simply the localization of RNAs. This is important because it shows how sequence-encoded features in RNAs can drive the formation of specific scaffolds that drive higher-order condensates. Given this novelty, I consider this paper worth of publication in Nature Cell Biology following revisions.

Major Concern:

The most important gap in the story in my opinion is in the characterization of the UU mutant oskar RNA rescue, particularly with regards to translation of the resulting oskar protein and oskar RNA levels. The Ephrussi group previously reported Bruno is a translational repressor of oskar.

[https://www.cell.com/fulltext/S0092-8674\(06\)00129-2](https://www.cell.com/fulltext/S0092-8674(06)00129-2) Given the reported reduction of Bruno RNA binding affinity as shown in figure 2C which differs from the cell free data (Figure 3), this suggest that Bruno by itself does not bind strongly in the mutated region and I am wondering if there could be some sort of cooperative binding with another protein in vivo. I am curious to see how the translation products of the two rescue sequences compare with something like a western blot or IF at different

development stages 2. Is there a difference in translation also via luciferase reporter as done in the previous cell paper noted above? This is because although the authors do not see significantly abrogated binding, they do see significant condensation difference (Figure 3). It would add to the understanding of the functional roles of the condensates to see whether oskar translation is being altered by condensation or not, as a purported role for condensates is the regulation of translation. I would say if for some reason some of these translation tests cannot be performed, I would still support publication but encourage the authors to attempt to assess protein products to better understand how different scales of condensates work.

Minor concerns:

Figure 1 B: please show ladder to demonstrate Oskar is actually a dimer.

Consider listing the components of “zero salt buffer” in the figure

Would a better title be granule “coarsening” as small assemblies of Bruno and oskar are still formed in the mutant case.

Figure 2a and 2b please show ladder.

Figure 2d Oskar protein reportedly binds its own mRNA

<https://www.pnas.org/doi/full/10.1073/pnas.1515568112> therefore it would probably worth blotting for wildtype oskar protein in the extracts to see if this is mediated by the UU mutation.

Figure 3b and C please show ladders.

Figure 3E How representative are the concentrations tested to the in vivo setting? Consider a phase diagram or more rationale for the conditions reported.

Figure 3F are the UU and the 292 mutants significantly different?

Was the labeling ratio of the different RNAs similar? Consider also quantifying the protein signal to assess how the ratio of protein to RNA is impacted.

Figure 5 Do the UU mutants show reduced fertility particularly under starvation conditions?

Supplemental 3a include the ladder. Are you confident the band annotation is correct given the smaller than monomer band present in the gel? Is this a degradation product or a contamination in the protein prep? Consider running a protein alone lane.

AUTHOR CONTRIBUTIONS – must be included after the Acknowledgements, detailing the contributions

of each author to the paper (e.g. experimental work, project planning, data analysis etc.). Each author should be listed by his/her initials.

Methods should be written concisely, but should contain all elements necessary to allow interpretation and replication of the results. As a guideline, Methods sections typically do not exceed 3,000 words. The Methods should be divided into subsections listing reagents and techniques. When citing previous methods, accurate references should be provided and any alterations should be noted. Information must be provided about: antibody dilutions, company names, catalogue numbers and clone numbers for monoclonal antibodies; sequences of RNAi and cDNA probes/primers or company names and catalogue numbers if reagents are commercial; cell line names, sources and information on cell line identity and authentication. Animal studies and experiments involving human subjects must be reported in detail, identifying the committees approving the protocols. For studies involving human subjects/samples, a statement must be included confirming that informed consent was obtained. Statistical analyses and information on the reproducibility of experimental results should be provided in a section titled “Statistics and Reproducibility”.

All Nature Cell Biology manuscripts submitted on or after March 21 2016 must include a Data availability

statement as a separate section after Methods but before references, under the heading "Data Availability". . For Springer Nature policies on data availability see <http://www.nature.com/authors/policies/availability.html>; for more information on this particular policy see <http://www.nature.com/authors/policies/data/data-availability-statements-data-citations.pdf>. The Data availability statement should include:

- Accession codes for primary datasets (generated during the study under consideration and designated as "primary accessions") and secondary datasets (published datasets reanalysed during the study under consideration, designated as "referenced accessions"). For primary accessions data should be made public to coincide with publication of the manuscript. A list of data types for which submission to community-endorsed public repositories is mandated (including sequence, structure, microarray, deep sequencing data) can be found here <http://www.nature.com/authors/policies/availability.html#data>.
- Unique identifiers (accession codes, DOIs or other unique persistent identifier) and hyperlinks for datasets deposited in an approved repository, but for which data deposition is not mandated (see here for details <http://www.nature.com/sdata/data-policies/repositories>).
- At a minimum, please include a statement confirming that all relevant data are available from the authors, and/or are included with the manuscript (e.g. as source data or supplementary information), listing which data are included (e.g. by figure panels and data types) and mentioning any restrictions on availability.
- If a dataset has a Digital Object Identifier (DOI) as its unique identifier, we strongly encourage including this in the Reference list and citing the dataset in the Methods.

We recommend that you upload the step-by-step protocols used in this manuscript to the Protocol Exchange. More details can found at www.nature.com/protocolexchange/about.

All imaging data should be accompanied by scale bars, which should be defined in the legend.

Cropped images of gels/blots are acceptable, but need to be accompanied by size markers, and to retain visible background signal within the linear range (i.e. should not be saturated). The boundaries of panels with low background have to be demarked with black lines. Splicing of panels should only be considered if unavoidable, and must be clearly marked on the figure, and noted in the legend with a statement on whether the samples were obtained and processed simultaneously. Quantitative comparisons between samples on different gels/blots are discouraged; if this is unavoidable, it should only be performed for samples derived from the same experiment with gels/blots were processed in parallel, which needs to be stated in the legend.

Unprocessed scans of all key data generated through electrophoretic separation techniques need to be presented in a supplementary figure that should be labelled and numbered as the final supplementary figure, and should be mentioned in every relevant figure legend. This figure does not count towards the total number of figures and is the only figure that can be displayed over multiple pages, but should be provided as a single file, in PDF or TIFF format. Data in this figure can be displayed in a relatively informal

style, but size markers and the figures panels corresponding to the presented data must be indicated.

The total number of Supplementary Figures (not including the “unprocessed scans” Supplementary Figure) should not exceed the number of main display items (figures and/or tables (see our Guide to Authors and March 2012 editorial <http://www.nature.com/ncb/authors/submit/index.html#suppinfo>; <http://www.nature.com/ncb/journal/v14/n3/index.html#ed>). No restrictions apply to Supplementary Tables or Videos, but we advise authors to be selective in including supplemental data.

GUIDELINES FOR EXPERIMENTAL AND STATISTICAL REPORTING

REPORTING REQUIREMENTS – We are trying to improve the quality of methods and statistics reporting in our papers. To that end, we are now asking authors to complete a reporting summary that collects information on experimental design and reagents. The Reporting Summary can be found here <https://www.nature.com/documents/nr-reporting-summary.pdf>) If you would like to reference the guidance text as you complete the template, please access these flattened versions at <http://www.nature.com/authors/policies/availability.html>.

Author Rebuttal to Initial comments

Response to the Reviewers' comments

We are very grateful to the reviewers for evaluating our work and thank them for their insightful and constructive comments. In our revised manuscript, we have included three additional Extended Data figures and highlighted all text changes in blue. In brief, following the reviewers' suggestions to test if the RNA kissing-loop interaction was sufficient for condensation, we tested if a compensatory mutation that restores base pairing with the UU mutant could rescue condensation. We further assessed the effect of varying the base-pairing strength of the kissing-loop interaction in scaffolding condensation. These experiments demonstrate not only that the kissing loop RNA-RNA interaction is essential for Bruno-driven phase separation, but also that the strength of the interaction modulates the extent of condensation. We also assessed the specific effect of the UU mutation on translational regulation of *oskar* RNA and observed that disruption of the kissing-loop interaction impairs spatial regulation of *oskar* translation. We have expanded our discussion of the emerging role of stable, sequence-specific RNA-RNA interactions on condensation and the impact of condensate assembly. All suggested additional control experiments, replicates for quantification and statistics have been performed and included in the revised manuscript. Unprocessed images of all gels/blots and a Supplementary Table including all numerical source data are provided. Based on a suggestion of Reviewer 3, we have modified the title of the manuscript to 'An architectural role of specific RNA-RNA interactions in *oskar* granules' which we believe more accurately describes our findings.

Reviewer #1:

Remarks to the Author:

It has been known that relatively weak, multivalent, and often promiscuous interactions between RNA-RNA, RNA-RBP, and IDR/PrID in RBPs promote the formation of biomolecular condensates. In this manuscript, Bose et al. report that a stable base-pairing interaction between the kissing loops of the SL2b in the *osk* RNA contributes to mesoscale granule formation in the *Drosophila* oocyte. The disruption of the kissing-loop interaction showed defects in *osk* RNA localization to the posterior pole of the oocyte. This *osk*-UU mutant RNA did not form mesoscale granules. The authors also showed that the 359-base

osk RNA fragment containing the SL2b and Bruno (Bru) binding sequence pulled down Bru from ovary lysates. Notably, the strength of the interaction was dependent on RNA dimerization, which was also important for Bru-mediated condensation in both in vivo and in vitro. Furthermore, the authors showed that the kissing-loop interaction of osk RNA was crucial for the stress-induced formation of P-bodies that contain osk RNA and Bru. The authors propose that the sequence-specific intramolecular RNA-RNA interaction acts with IDR-containing RBPs in driving condensation.

A clear demonstration of the contribution of the specific base-pairing interaction between RNA molecules in phase separation is, to my knowledge, novel and should be a significant finding. A broad audience in developmental and molecular cell biology including many readers of Nature Cell Biology will be interested in this discovery, which will stimulate related research fields. Thus, the manuscript should be worthy of reporting in premiere-class journals.

The manuscript is well-written and easy to follow the authors' logical flow. Qualities of figures are generally high. I think that statistical processing is adequately operated.

However, it remains unclear from the current resolution of data whether the kissing-loop interaction of osk mRNA directly stimulates condensation, or the dimerization generates a new platform for a specific client RBP, which in turn contributes to condensation. Although the novelty of their finding will be unchanged in either case, I think that the authors need to discriminate between these two possibilities and describe a clearer picture of how a stable base-pairing interaction contributes to condensation. I would like to suggest several possible experiments below. As this field of research has been extremely complicated, there may be misunderstandings in my comments below. If the authors find such statements, please rebuttal to my suggestions.

We are pleased the reviewer appreciated the novelty and significance of our study addressing the *in vivo*, functional consequences of specific RNA-RNA interactions in granule assembly and thank them for their suggestions. Below we address the reviewer's specific concerns. In brief, while it is difficult to prove unambiguously, we do not believe that *oskar* RNA dimerization *per se* stimulates condensation, but rather that this interaction is important in stabilizing the ribonucleoprotein network that forms the platform for condensate assembly.

Major concerns:

1. The authors showed that the osk-UU mutant did not dimerize, failed to form larger granules in the oocyte, and showed defects in its posterior localization. These results indicate the necessity of the kissing-loop interaction for proper osk RNA behaviors. I wonder if the dimerization through the kissing loops in the SL2b is sufficient for osk RNA behaviors. That means, the authors could provide data on the restoration of the kissing-loop interaction can rescue the defects. The authors' group previously reported that the AA mutation in the loop acted as a compensatory mutant and restored the dimerization with the UU-loop (Jambor et al. RNA 2011). The authors can examine whether co-expression of the AA form of the transgene can rescue defects observed in the osk-UU mutant. Also, the authors

could conduct whether the presence of both *osk-UU359* and *osk-AA359* RNAs promotes Bru-mediated condensation *in vitro*.

We thank the reviewer for this suggestion, which led us to perform the following experiments: We first examined rescue of *oskar* dimerization *in vitro* using the compensatory AA mutant¹ (RR Fig. 1a and Extended Data Fig. 6a-b). When UU₃₅₉ and AA₃₅₉ RNAs were mixed at 1:1 molar ratio, there was no clear increase in the dimeric form (fold rescue = 1.07 ± 0.047 ; n=3). Addition of Bruno resulted in a small but significant increase in condensate formation (Extended Data Fig. 6a-b). Given these results, we did not expect a major rescue of *oskar* function upon co-expression of *oskarAA* and *oskarUU* *in vivo*.

Nevertheless, to examine rescue *in vivo*, we expressed solely the *oskar* 3'UTR harbouring the AA mutation, as expression of the genomic *oskarAA* (including the coding region) would lead to overproduction of Oskar protein and confound the effect on translation. It should be noted, however, that expression of the 3'UTR-AA RNA in the *osk-UU* mutant leads to an overall increase in *oskar* 3'UTR levels in the system and alter the ratio of *oskar* 3'UTR to Bruno, which would itself affect granule formation. smFISH staining revealed that posterior localization of *oskar* was not rescued; the cortical distribution of the RNA persisted in late oogenic stages in a manner identical to that of the UU mutants (RR Fig. 1b). No significant rescue in partitioning of *oskar* RNA into granules was observed (RR Fig. 1c and Extended Data Fig. 6c), and ectopic translation of Oskar protein in the oocytes and early embryos persisted, suggesting that the translation deregulation observed in the non-dimerizing UU mutant (Extended Data Fig. 3) could not be rescued (RR Fig. 1d). The data on granule assembly *in vitro* and *oskar* partitioning *in vivo* (RR Fig. 1c) are now included in new Extended Data Fig. 6 in the revised manuscript.

These experiments show that, overall, complementation with AA mutation could not restore the kissing-loop interaction and could therefore not rescue the defects. These experiments however also raised the interesting possibility that lack of rescue may be a result of the lower base-pairing strength of A-U in comparison to G-C present in the WT *oskar* sequence (discussed in more detail in response to point 2).

RR Fig. 1. a, Schematic representation of the kissing loop palindromic sequence showing the degree of base pairing in WT, UU and AA+UU. **b**, smFISH staining for *oskar* RNA in the indicated genotypes showing the defects in posterior localization of *oskar* RNA. Note that the cortical accumulation of *oskar* in the case of *oskar*UU could not be rescued by the compensatory AA mutation. **c-d**, Granule formation and partitioning of *oskar* RNA into granules (c), as well as ectopic accumulation of Oskar protein in oocytes and early embryos (d), could not be rescued upon co-expression of the *oskar* 3'UTR bearing the compensatory AA mutation in *oskar* UU egg chambers. Error bars represent SD; Unpaired Student's t-tests were used for comparisons; ns = non-significant.

2. If the dimerization of *osk* RNA through the kissing-loop interaction is sufficient to promote condensation independently from the face-to-face duplication of the SL2b stem, the addition of an engineered kissing-loop sequence (lines 320-322) in the *osk*-UU RNA might rescue defects. I think that the authors could conduct these types of experiments.

Inspired by the reviewer's suggestion, we used our *in vitro* setup and replaced the hexanucleotide palindromic sequence of the loop with palindromic sequences of varying base pairing strengths. We chose the Dimerization Initiation Site (DIS) in the 5' region of the HIV type 1 genomic RNA harbouring a stem loop (SL1) which mediates dimerization of the genomic RNA via a kissing loop interaction^{2,3}. Experiments have demonstrated that the central two G-C base pairs within the GC-rich palindrome are critical for stability of the kissing loop, and mutants containing more than one A-U base pair replicated poorly⁴⁻⁶. Therefore, we replaced the WT hexanucleotide sequence with palindromic sequences of two isotopes of HIV, which we named HIV-1 (GC content = 66.7%) and HIV-2 (GC content = 100%) (new panel Fig. 3g in the revision). The remaining sequence of the *oskar* 359 RNA was unaltered. The HIV-2 sequence, with higher GC content, was predominantly dimeric and formed higher order oligomers, while the HIV-1 sequence, with lower GC content, was significantly less

dimeric (Fig. 3g). Addition of Bruno led to significantly higher condensate formation in case of the HIV-2 RNA compared to the HIV-1, suggesting that stronger kissing loop interactions contribute to condensate formation in our minimal system (Fig. 3h, i). This also explains why the compensatory AA mutation could only weakly rescue condensation, owing to the lower strength of the A-U base pairing compared to G-C (Extended Data Fig. 6).

It was tempting to test the effect of the varying strengths of kissing loop interactions *in vivo* in the context of *oskar* function. However we noted that although both HIV sequences promoted condensation with Bruno *in vitro* to a greater extent than did the UU mutant (also together with the compensatory AA mutation), even the HIV-2 sequence (GCGCGC), which is essentially a scrambled version of the WT *oskar* (CCGCGG), could not rescue higher-order RNA oligomerization and condensate formation to the same extent as the WT. This suggests that not only the base composition, but also the nucleotide arrangement/geometry determines the strength of the kissing loop interaction. Therefore, as with the *oskar* AA compensatory mutation (Extended Data Fig. 6), it is unlikely that replacement of the WT palindrome with HIV-1 or HIV-2 sequences *in vivo* would fully rescue *oskar* function. Considering this and the fact that this approach would involve generation of transgenic fly lines and extensive optimization (e.g. of levels of expression), we chose not to focus our efforts on testing *in vivo*.

Based on these results we propose that stereospecific RNA-RBP interactions seed RNPs, which by virtue of multivalent interactions between both the protein and RNA scaffold molecules can recruit additional scaffold molecules⁷ and enrich client proteins to establish the RNP granule network. In the case of the UU mutant (reported here) or upon removal of Bruno PrLD⁸, the granule scaffolding itself is abrogated, and consequently the partitioning of client proteins is affected. The above *in vitro* experiments using synthetic RNA constructs add to the evidence that RNA dimerization via the kissing loop interaction is essential for scaffolding condensate assembly. Fine tuning of the properties of the kissing loop interaction modulates condensation. Although we can only speculate as to whether dimer formation generates a new interaction platform for a specific client RBP, our study emphasizes that the observed effects stem from the upstream defect in the RNA scaffold molecule, which leads to a cascade of defects in granule function.

3. In an *in vitro* assay, conditions of the reaction (components, concentrations, salts, temperature, etc.) can be optimized, and do not reflect *in vivo* situations. For example, in the reaction shown in Fig 3e, the addition of an RBP that interacts with SL2b might strongly stimulate condensation. Notably, the authors showed that the association of Staufen (Stau) to the *osk*359 RNA fragment was reduced in the UU mutant; the reduction level was similar to that observed for Bruno (Extended Data Fig. 2). Since Stau contains multiple dsRNA-binding domains (Ramos et al. EMBO J. 2000), it could interact with the stem structure in the SL2b (as has been predicted in Mohr et al. PLOS Genet. 2021) and the interaction might be enhanced when SL2b stems become tandemly oriented by the kissing-loop interaction.

The authors showed that the *osk*-UU RNA failed to localize to the posterior pole of the oocyte (Fig 1c). Given that Stau is crucial for the posterior localization of *osk* RNA in the oocyte, the defects might be caused by the reduced interaction between *osk* RNA and Stau (the idea is also supported by data shown in Extended Data Fig. 2). A large fraction of the *osk*-UU RNA also remained at the anterior and lateral cortex in the oocyte (Fig 1c), which

suggests that the *osk*-UU RNP still associates with the active form of dynein-mediated transport machinery even in the oocyte. It has been reported that a dynein machinery component Egalitarian (Egl) antagonizes with Stau for localization of *osk* RNA within the oocyte (Mohr et al. PLOS Genet. 2021). The authors' group also recently reported an antagonized relationship between Stau and Egl (Gáspár et al. J. Cell Biol. 2023). Furthermore, the authors' group has reported that Egl is a candidate RBP that interacts with the SL2b stem for dynein-mediated transport (Jambor et al. RNA 2014). These observations additionally support the idea that the defects observed in the *osk*-UU mutant are caused by the reduction of the *osk* RNA-Stau interaction. In this scenario, the kissing-loop interaction strengthens the interaction between the tandemly oriented SL2b stems and Stau, leading to the displacement of Egl from each stem. I think that the authors could examine and discuss the potential roles of Stau and Egl in SL2b-mediated condensation and posterior localization of *osk* RNA. For example, the authors could test whether Stau interacts with the SL2b stems, especially with the face-to-face duplicated form. The authors could also examine whether the association of Egl to the *osk*³⁵⁹ RNA in the pull-down assay (Fig 2d) is stronger in the *osk*-UU mutant RNA (due to lack of Stau-mediated interference).

Staufen is an RBP which associates with *oskar* RNPs exclusively in the oocyte compartment⁹ and is reported to promote kinesin-mediated transport of *oskar* to the posterior pole by antagonising the Dynein-based transport machinery^{10,11}. However, the role of Staufen in phase separation of *oskar* transport granules is indeed unexplored.

We observed a reduction in Staufen association in the case of the UU mutant in our *ex vivo* pull down assays (Fig. 2D and Extended Data Fig. 4a). However, EMSAs with purified Staufen protein and OES67 (WT and UU) showed that both WT and UU OES bind Staufen and form RNP complexes (new panel Extended Data Fig. 4b in the revision). Thus, OES dimerization is not essential for Staufen binding *in vitro* and the observed reduction in Staufen association (Fig. 2D) with *oskar* UU is possibly more likely to be due to impaired granule scaffolding. Our previous data with Bruno Δ N show that in absence of Bruno-driven phase separation, granule assembly is abrogated even in presence of wild-type *oskar* RNA⁸. In this scenario, although the kissing loop interaction is not affected, posterior localization of *oskar* is still disrupted in the oocyte. Therefore, granule assembly is upstream of granule localization and it is unlikely that reduction of Staufen association with *oskar* is responsible for the observed UU phenotype.

We additionally examined Egl association in the *ex vivo* pull down experiments (Fig. 2D and Extended Data Fig. 4a). Recruitment of Staufen to *oskar* RNPs in the oocyte is known to lead to dissociation of the Dynein adaptor protein, Egalitarian (Egl)^{10,11}. Therefore, reduced Staufen interaction in the case of *oskar* UU is expected to lead to higher Egl retention. We instead observed that Egl association is not strengthened, but is rather slightly reduced in the case of UU, suggesting that the localization defects do not arise from defective interplay between the two RBPs Staufen and Egl.

Minor points:

1. The authors show that the *osk*-UU RNA does not form large granules (Fig 1e). I wonder if the wild-type *osk* RNA forms granules in *stau* mutant oocytes.

To address this question, we knocked down Staufen specifically in the germline (RR Fig. 2a). We observed that *oskar* localization is affected in Staufen RNAi egg chambers (RR Fig. 2b), as reported previously¹¹. Examination of the *oskar* granules revealed a slight reduction in the partition coefficient of *oskar* RNA in granules (RR Fig. 2c, d). Since Staufen enriches into the granules exclusively in the oocyte compartment as a client protein, it is possible that Staufen might have a yet unidentified role in *oskar* granule remodelling, independent of Bruno and the kissing loop interaction.

Since we do not systematically study all the different client proteins in this manuscript, we chose not to include the Staufen data in the revised manuscript as it would dilute the focus of the study.

RR Fig. 2. **a**, Reduction of Staufen protein in the ovaries upon RNAi mediated knock down as detected by western blotting. Wild type: *w¹¹¹⁸*. **b**, smFISH showing mislocalization of *oskar* RNA in *stau* RNAi egg chambers. **c-d**, *oskar* granule assembly is slightly impaired upon Staufen knock down as evident from the histogram of pixel intensities of the *oskar* channel

(c) and the significant reduction in partitioning of *oskar* into granules (d). Error bars represent SD. Unpaired Student's t-tests were used for comparisons. Significance level: * ≤ 0.05 .

2. Lines 107-110: The authors argue that the polarity in the oocyte is intact in the *osk*-UU mutant by examining *grk* RNA localization to the anterior-dorsal corner of the oocyte (Extended Data Fig. 1). Considering the microtubule organization and *osk* RNP behaviors within the oocyte (Zimyanin et al Cell 2008; Parton et al. J. Cell Biol. 2011), the data on *grk* RNA localization alone is insufficient to judge the polarity in the oocyte. I think that the authors will have to examine the posterior enrichment of Kinesin (either by endogenous protein detection or using the kinesin motor domain- β -galactosidase marker) or directly examine microtubule organization (e.g. EB1-GFP imaging).

Considering the already complex genotypes of the flies, which harbor transgenes and/or mutations on both of the second and third chromosomes, introducing the EB1-GFP transgene was very difficult. Therefore, we examined microtubule organization directly in the mid-oogenesis oocytes by immunostaining using a FITC coupled anti-alpha-Tubulin antibody. As in *oskar* WT oocytes, *oskar* UU oocytes formed an anterior-to-posterior microtubule gradient, with a high density of microtubules originating from the anterior cortex and a specific depletion at the posterior pole. In contrast, this organization was completely disrupted in *gurken* mutant oocytes (*grk* 2B6/2E2), where microtubules nucleated from all around the oocyte cortex and were depleted from the center of the oocyte. These data confirms the known mispolarization of the microtubule network in *gurken* mutants¹² and are now included in the revised Extended Data Fig. 2b.

The clear anterior-posterior microtubule gradient, along with the correct anterodorsal localization of *gurken* RNA and of the oocyte nucleus (Extended Data Fig. 2a), strongly suggest that oocyte polarity is not substantially affected by the *oskar* UU mutation.

3. Extended Data Fig. 2: I suggest that the authors should conduct the experiment more than triplicates and conduct statistical analysis.

We have performed more replicates of the RNA affinity capture experiment for statistical quantification, and probed for additional *oskar*-associated RBPs. The quantifications are now included in the revised manuscript (Fig. 2d and Extended Data Fig. 4a).

4. The authors previously reported that the *osk* RNP is a solid-like condensate (Bose et al. Cell 2022). Does a relatively strong and stable base-pairing interaction between the kissing loops of the *osk* RNA contribute to its solid-like status?

I think that many researchers tackling the condensation issues feel that stable RNA-RNA interactions provide a negative effect driving condensation. If possible, I think it would be wonderful if the authors could discuss this issue.

We thank the reviewer for bringing up this point. While the contribution of intermolecular RNA-RNA interactions in LLPS is acknowledged, the nature of these interactions (sequence-specific tertiary interactions and their strength vs promiscuous base pairing) is explored to a much lower extent. Any two RNAs have the potential to interact over short stretches by virtue

of degenerate base-pairing, especially at high RNA concentrations. Such interactions have been shown to drive LLPS either of RNA on its own¹³⁻¹⁶ or in collaboration with RNA-binding proteins^{17,18}. As we have discussed in the original submission, one example of sequence-specific RNA-RNA interactions in condensation comes from the filamentous fungus *Ashbya gossypii* where co-assembling *SPA2* and *BNI1* mRNAs enrich in apically localized granules, whereas self-assembling *CLN3* mRNA forms distinct condensates around nuclei¹⁹. Interestingly, alteration of the secondary structure of *CLN3* leads to co-packaging of *CLN3* with the *SPA2/BNI1* granules, emphasizing the importance of RNA sequence and structure-dependent interactions in specifying condensate composition. These condensates exhibit either liquid-like or gel-like behaviour depending on the protein-to-RNA ratio, indicating that stable, sequence-specific RNA-RNA interactions do not necessarily determine the material state of these assemblies. We note however that the authors did not test the strength of these RNA-RNA interactions. Our study provides a clear example of a positive role of specific and stable RNA-RNA interactions in scaffolding condensation. Following the reviewer's suggestion, we now discuss this in the revised manuscript (lines 399-408).

It is interesting to speculate if varying RNA-RNA interaction strength can contribute to different condensate material properties. In fact, a very recent study showed that short RNAs involved in repeat-expansion disorders can undergo liquid-to-solid phase transition *in vitro* via stable G-quadruplex structure formation²⁰. Our previous report showed that *oskar* transport granules are solid-like condensates and that RBPs such as Bruno, Hrp48 contribute to the solid-like material properties⁸. Moreover, we reported that the material properties of the granules with WT *oskar* RNA can be modulated from a solid to a more liquid-like state by tethering of FUS LC to *oskar* RNA or by depletion of Hrp48. However, whether the strong base-pairing interactions within the kissing loop also contribute to the solid-like material properties of *oskar* granules is difficult to test, as *oskar* UU does not form condensates *in vitro* or *in vivo* (Figs.1 and 3).

Reviewer #2:

Remarks to the Author:

In this manuscript, the authors explore the role of a palindromic sequence, called the kissing-loop, located within the *oskar* mRNA 3'UTR in mediating RNA-protein granule assembly. Through a combination of *in vitro* RNA biochemistry and *in vivo* cell biology, they discover that the kissing-loop helps to promote dimerization of *oskar* mRNA and ultimately drive the formation of higher order ribonucleoprotein assemblies containing the RNA binding protein Bruno. Additionally, they show that the *oskar* mRNA, and specifically the kissing-loop, is required for the formation of P bodies upon nutrient deprivation. The study provides important and timely insight into the role of specific intermolecular RNA-RNA interactions in bimolecular condensate formation and as such will likely appeal to a broad audience. The experiments are sound and the conclusions are largely supported by the data (see comments below).

We thank the reviewer for appreciating the importance of our findings on the functional consequences of intermolecular RNA-RNA interactions in condensate formation.

Specific Comments

Is it possible that protein interactions with the kissing-loop underly its role in granule assembly, rather than RNA dimerization? Presumably not Bruno based on your results but possibly other RBPs, such as Me31B. One way to test this would be to alter the nucleotide sequence of the kissing-loop domain in a way that doesn't substantially impact RNA dimerization.

We thank the reviewer for the question and appreciate the concern. Our *in vitro* condensate assembly experiments using the scaffold protein Bruno and WT/UU RNAs (Figs. 3d-f) show that the kissing loop interaction is crucial for condensate formation in a minimal system, highlighting the role of specific RNA-RNA interactions in scaffolding granule assembly, without the involvement of a kissing-loop interacting protein (note that Bruno binding sites is located elsewhere in the *oskar* mRNA 3'UTR sequence, Figs. 1a, 2a). In order to further substantiate the causal role of the RNA-RNA interactions, we now used two approaches:

1. Introduction of a compensatory mutation that restores base-pairing with UU:

Using the compensatory AA mutant¹ (RR Fig. 1 above and new Extended Data Fig. 6a-b in the revision), we could not significantly rescue dimerization with UU. Addition of Bruno resulted in a small but significant increase in condensate formation (Extended Data Fig. 6a-b). Given the weak effect of the AA compensatory mutation on dimerization, we did not expect a significant rescue of *oskar* function *in vivo*. Nevertheless, to experimentally test this, we expressed the *oskar* 3'UTR alone harbouring the AA mutation, since expression of the genomic *oskar*AA will lead to over-production of Oskar protein thereby confounding the effect on translation. It should however be noted that expression of the 3'UTR-AA RNA in UU mutant will lead to an overall increase in *oskar* 3'UTR levels in the system and alter the ratio of *oskar* 3'UTR to Bruno, which would itself affect granule formation. Co-expression of AA with UU indeed did not rescue granule formation and *oskar* localization *in vivo* (new Extended Data Fig. 6c). Translation deregulation and ectopic Oskar protein accumulation observed in UU mutants (new Extended Data Fig. 3) also persisted upon AA co-expression similar to the UU mutant alone. We reasoned that the lack of rescue in the above condition is due to the low base-pairing strength of A-U compared to that of G-C present in the WT *oskar* sequence.

2. Analysing the strength of the kissing loop interaction on condensate formation:

To address the effect of the strength of the kissing loop interaction on condensate assembly, we used our *in vitro* setup and replaced the hexanucleotide palindromic sequence of the loop with palindromic sequences of varying base pairing strengths. We chose the Dimerization Initiation Site (DIS) in the 5' region of the HIV type 1 genomic RNA harbouring a stem loop (SL1) which mediates dimerization of the genomic RNA via a kissing loop interaction^{2,3}. Experiments have demonstrated that the central two G-C base pairs within the GC-rich palindrome are critical for stability of the kissing loop, and mutants containing more than one A-U base pair replicated poorly⁴⁻⁶. Therefore, we replaced the WT hexanucleotide sequence with palindromic sequences of two isotypes of HIV, which we named HIV-1 (GC content = 66.7%) and HIV-2 (GC content = 100%) (new panel Fig. 3g in the revision). The remaining

sequence of the *oskar* 359 RNA was unaltered. The HIV-2 sequence, with higher GC content, was predominantly dimeric and formed higher order oligomers, while the HIV-1 sequence, with lower GC content, was significantly less dimeric (Fig. 3g). Addition of Bruno led to significantly higher condensate formation in case of the HIV-2 RNA compared to the HIV-1, suggesting that stronger kissing loop interactions contribute to condensate formation in our minimal system (Fig. 3h, i). This also explains why the compensatory AA mutation could only weakly rescue condensation, owing to the lower strength of the A-U base pairing compared to G-C (Extended Data Fig. 6).

It was tempting to test the effect of the varying strengths of kissing loop interactions *in vivo* in the context of *oskar* function. However we noted that although both HIV sequences promoted condensation with Bruno *in vitro* to a greater extent than did the UU mutant (also together with the compensatory AA mutation), even the HIV-2 sequence (GCGCGC), which is essentially a scrambled version of the WT *oskar* (CCGCGG), could not rescue higher-order RNA oligomerization and condensate formation to the same extent as the WT. This suggests that not only the base composition, but also the nucleotide arrangement/geometry determines the strength of the kissing loop interaction. Therefore, as with the *oskar* AA compensatory mutation (Extended Data Fig. 6), it is unlikely that replacement of the WT palindrome with HIV-1 or HIV-2 sequences *in vivo* would fully rescue *oskar* function. Considering this and the fact that this approach would involve generation of transgenic fly lines and extensive optimization (e.g. of levels of expression), we chose not to focus our efforts on testing *in vivo*.

The above *in vitro* experiments using synthetic RNA constructs add to the evidence that RNA dimerization via the kissing loop interaction is essential for scaffolding condensate assembly. Fine tuning of the properties of the kissing loop interaction modulates condensation. Based on these results we propose that stereospecific RNA-RBP interactions seed RNPs, which by virtue of multivalent interactions between both the protein and RNA scaffold molecules can recruit additional scaffold molecules⁷ and enrich client proteins to establish the RNP granule network. In the case of the UU mutant (reported here) or upon removal of Bruno PrLD⁸, the granule scaffolding itself is abrogated, and consequently the partitioning of client proteins is affected. Accordingly, we reported a general reduction in *oskar*-associated RBPs in our *ex vivo* RNA affinity capture experiment for the UU mutant. We now also specifically tested association of Me31B, as suggested by the reviewer, and observed a slight reduction in Me31B recruitment in the case of UU (RR Fig. 3). We observed a significant reduction in association of client protein Staufen (Fig. 2d and Extended Data Fig. 4).

Although we can only speculate as to whether dimer formation generates a new interaction platform for a specific client RBP, our study emphasizes that the observed effects stem from the upstream defect in the RNA scaffold molecule, which leads to a cascade of defects in granule function.

RR Fig. 3: RNA affinity capture experiment (described in Fig. 2d) using WT₃₅₉ and UU₃₅₉ RNA, probed for the association of Me31B protein. Beads without RNA serve as a negative control.

The relative level of the respective RNAs (determined by qPCR) pulled down by the streptavidin beads were used to normalize the amount of Me31B pulled down.

Fig. 1b, 2a, 2b, 3b, 3c. Size markers in the gel images would be useful.

We apologise for this omission. We thank the reviewer for pointing this out. Since in all our experiments we used atto633-labelled RNAs for higher sensitivity of detection (far-red channel), we usually omitted the RNA marker (green channel).

Following the reviewer's comment, We have now re-run all the RNAs used in this study with size markers (two-channel gel images now shown in Extended Data Fig. 1). Based on these reference data, we now indicate the size markers in figures 1, 2 and 3.

Fig. 2d. Why is there no variation in wt in the bar plot? Presumably this is related to normalization but the variation should nonetheless be displayed (standard deviation from the mean) and of course the statistical analysis should also take into account variation in wt.

We thank the reviewer for their comment and apologize for the lack of clarity.

For each replicate, we first quantified the relative levels of the WT and UU RNAs (setting WT to 1). The obtained factor was then used to normalize the amount of Bruno pulled down by UU. Note that these are not absolute values. Therefore, the error bar on UU represents the variation in 'relative enrichment' of Bruno between WT and UU in the different experimental replicates. It does not reflect the inherent variation within one sample type. We now explain this in the figure legend of the revised manuscript.

Fig. 5b and associated results. In concluding that Bruno localizes to P bodies, you don't show co-localization of Bruno with typical P body markers. Presumably the logic here is that because *oskar* overlaps with Me13B and Bruno overlaps with *oskar*, that Bruno is also in P bodies?

Indeed. Since *oskar* overlaps with the P-body marker protein Me31B and Bruno colocalizes with *oskar*, it is expected that Bruno and Me31B also colocalize. Nevertheless, to confirm this, we performed immunodetection of Me31B on Bruno-KI EGFP flies after starvation. The data showing localization of Bruno to P-bodies have been included in the Extended Data Fig. 7 of the revised manuscript.

Reviewer #3:

Remarks to the Author:

In the paper entitled, "An architectural role of *oskar* mRNA in granule assembly" Bose et al propose that RNA oligomerization dictated by the kissing stem loops of *oskar* mRNA is responsible for 2 varieties of condensate formation and *oskar* RNA localization. To my

knowledge this represents the first conclusive role for RNA oligomerization in promoting condensation, not simply the localization of RNAs. This is important because it shows how sequence-encoded features in RNAs can drive the formation of specific scaffolds that drive higher-order condensates. Given this novelty, I consider this paper worth of publication in Nature Cell Biology following revisions.

We thank the reviewer for appreciating the novelty of our findings and considering that it represents a substantial advance in the field with respect to the role of sequence-specific RNA-RNA interactions in driving condensation.

Major Concern:

The most important gap in the story in my opinion is in the characterization of the UU mutant oskar RNA rescue, particularly with regards to translation of the resulting oskar protein and oskar RNA levels. The Ephrussi group previously reported Bruno is a translational repressor of oskar. [https://www.cell.com/fulltext/S0092-8674\(06\)00129-2](https://www.cell.com/fulltext/S0092-8674(06)00129-2) Given the reported reduction of Bruno RNA binding affinity as shown in figure 2C which differs from the cell free data (Figure 3), this suggest that Bruno by itself does not bind strongly in the mutated region and I am wondering if there could be some sort of cooperative binding with another protein in vivo. I am curious to see how the translation products of the two rescue sequences compare with something like a western blot or IF at different development stages 2. Is there a difference in translation also via luciferase reporter as done in the previous cell paper noted above? This is because although the authors do not see significantly abrogated binding, they do see significant condensation difference (Figure 3). It would add to the understanding of the functional roles of the condensates to see whether oskar translation is being altered by condensation or not, as a purported role for condensates is the regulation of translation. I would say if for some reason some of these translation tests cannot be performed, I would still support publication but encourage the authors to attempt to assess protein products to better understand how different scales of condensates work.

Fig 2d shows the RNA affinity capture assay in which we observe that Bruno associates less with UU₃₅₉ than with WT₃₅₉ RNA. Reduced association was also observed for the client protein Staufén. Please note that this does not reflect a reduced binding of the proteins to the UU RNA *per se*, as evident from the EMSAs with Bruno (Fig. 3b) and Staufén (new Extended Data Fig. 4b). This, together with the fact that the Bruno binding site (BRE C) is present in both WT and UU RNAs, indicates that the reduced association of Bruno with the UU RNA in the capture assay (Fig. 2d) is due to impaired RNA oligomerization. We conclude that Bruno binding to *oskar* RNA is not disrupted in the non-dimerizing mutant, but rather higher-order oligomerization of the Bruno-RNA complex and thus granule formation is perturbed (lines 213-223).

We further thank the reviewer for their question, which we take as an opportunity to discuss the impact of granule formation on translation control of *oskar* RNA. Upon immunostaining for Oskar protein in *oskarUU* egg chambers we detected ectopic accumulation of Oskar protein mostly in the antero-dorsal corner of mid-oogenesis egg chambers as well as early embryos (new Extended Data Fig. 3a). This indicates that translation regulation of *oskar* RNA is

impaired when *oskar* transport granule formation is compromised by the UU mutation. Ectopic accumulation of Oskar protein persisted in early *oskar* UU embryos, with a predominant localization in a specific region along the anterodorsal surface (also observed upon *oskar* over-expression²¹). Deregulation of *oskar* translation had detrimental consequences for embryonic development. While no defects in segment formation were noted in *oskar*WT embryos, more than 98% of the *oskar*UU embryos failed to undergo proper segmentation, as evident from the cuticle phenotypes (new Extended Data Fig. 3b). The majority of the embryos exhibited strong ‘bicaudal’ phenotypes, including duplicated abdominal segments in the place of the head and ectopic filzkörper (fk) structures, reinforcing the impact of increased *oskar* gene dosage on abdominal patterning^{21,22}.

This suggests that proper scaffolding of granule formation and recruitment of translation regulators is key to the establishment of translation repression in the transport granules. Since disruption of the kissing loop interaction interferes with condensate formation, we suggest that condensate formation is essential for translation control of *oskar*.

Minor concerns:

Figure 1 B: please show ladder to demonstrate Oskar is actually a dimer.

We apologise for this omission. Since in all our experiments we used atto633-labelled RNAs for higher sensitivity of detection (far-red channel), we usually omitted the RNA marker (green channel). Following the reviewer’s comment, we have now re-run all the RNAs used in this study along with size markers (two-channel gel images now shown in new Extended Data Fig. 1). Based on these reference data, we now indicate the size markers in figures 1, 2 and 3.

Consider listing the components of “zero salt buffer” in the figure

We have now included the composition of all three buffers used (Zero salt, Medium salt and High salt) in the figure legends.

Would a better title be granule “coarsening” as small assemblies of Bruno and *oskar* are still formed in the mutant case.

Coarsening refers to the process of appearance of larger condensates with time. Coarsening can occur via fusion or coalescence of smaller droplets or through Ostwald ripening where larger droplets grow at the cost of smaller droplets which are inherently unstable and dissolve. We do not have data to conclusively distinguish initial granule assembly from coarsening of smaller granules to form larger ones. Therefore, we would refrain from using the term in the title of the manuscript.

However, inspired by the suggestion, we have edited our manuscript title to “An architectural role of specific RNA-RNA interactions in *oskar* granules”, which we believe more effectively reflects our findings.

Figure 2a and 2b please show ladder.

We apologise for the omission and have included the ladder in the revised figure panels.

Figure 2d Oskar protein reportedly binds its own mRNA

<https://www.pnas.org/doi/full/10.1073/pnas.1515568112> therefore it would probably worth blotting for wildtype oskar protein in the extracts to see if this is mediated by the UU mutation.

Although Yang et al., showed with *in vitro* experiments using bacterially expressed Osk-N and *oskar* 3'UTR, that Oskar protein binds its own 3'UTR, *oskar* RNA granules do not colocalize with Oskar protein granules in the oocyte. This segregation is maintained through embryogenesis and is shown to be essential for pole cell formation^{9, 23}.

Furthermore, in replicates of the *ex vivo* RNA Affinity Capture experiment we now report in the revision (Fig. 2d), we also probed for the short isoform of Oskar, whose OSK domain we have previously shown has the intrinsic capacity to bind RNA (Jeske et al., 2015). We detected no binding of Short Oskar to either WT or UU transcripts (RR Fig. 4).

RR Fig. 4: RNA affinity capture experiment using WT₃₅₉ and UU₃₅₉ RNA, probed for the association of Short Oskar protein. Beads without RNA serve as a negative control. The relative level of the respective RNAs (determined by qPCR) pulled down by the streptavidin beads are indicated below the blot.

Figure 3b and C please show ladders.

We apologise for the omission. The ladder is now indicated in the revised figure.

Figure 3E How representative are the concentrations tested to the *in vivo* setting? Consider a phase diagram or more rationale for the conditions reported.

We previously estimated the *in vivo* concentrations of *oskar* RNA and Bruno in the transport granules⁸ (Bruno ~ 0.8 μ M, *oskar* ~ 0.9 μ M). However, we have not estimated the total cytoplasmic concentration of Bruno or *oskar* in the egg chamber. *In vitro*, Bruno undergoes phase separation through self-association at 10 μ M⁸. Therefore, in our current *in vitro* experiments, we aimed to keep Bruno below the saturation concentration and closer to the physiological range, and thus chose an intermediate 5 μ M concentration. At this concentration, Bruno is soluble and its phase separation requires addition of RNA. Thus, the

effect of the RNA species in driving Bruno condensation could be tested. This rationale has been highlighted in the text (lines 261-265).

Figure 3F are the UU and the 292 mutants significantly different?

Was the labeling ratio of the different RNAs similar? Consider also quantifying the protein signal to assess how the ratio of protein to RNA is impacted.

We thank the reviewer for their questions and suggestion. Indeed, a difference in the degree of labelling of the different RNAs can confound interpretation. Therefore, as suggested, we have used the Bruno-EGFP channel to quantify the condensate size in all the *in vitro* condensate assembly experiments (Fig. 3 and Extended Data Fig. 6).

Figure 5 Do the UU mutants show reduced fertility particularly under starvation conditions?

We have performed egg laying experiments with *oskar* WT and *oskar* UU flies. The *oskar* UU eggs never hatch owing to the strong bicaudal phenotype of the embryos (new Extended Data Fig. 3). Therefore, it is impossible to determine what would have been the impact on fertility under starvation conditions.

Supplemental 3a include the ladder. Are you confident the band annotation is correct given the smaller than monomer band present in the gel? Is this a degradation product or a contamination in the protein prep? Consider running a protein alone lane.

We apologise for this omission and include the ladder in all revised figures.

We also thank the reviewer for questioning the smaller band. To determine the origin of the smaller band that appears only in the case of the full length *oskar* 3'UTR transcript (not with the *oskar*₃₅₉ RNA), we extracted the two bands from the gel and ran them separately after heat-denaturation (RR Fig. 5a). This confirmed that the smaller band is not a super-folded conformation of the full 3'UTR. To conclusively identify the RNA species, we next performed RNA sequencing and mapped the sequence to bases 1-308 of the 3'UTR (RR Fig. 5b). It likely results from a premature termination of the T7 polymerase during the *in vitro* transcription. Since this fragment is upstream of the OES and is absent from the 359 fragment used in the majority of our experiments, the presence of the small fragment does not affect our conclusions.

RR Fig. 5: **a**, Gel electrophoresis of full length *oskar* 3'UTR (*left*) showing the smaller RNA species (*left*). Extraction followed by denaturation and re-electrophoresis of the two fragments confirms that the smaller species is not a folding isoform of the 1kb 3'UTR (*right*). **b**, RNA sequencing revealed that the smaller fragment corresponds to nucleotides 1-308 of the *oskar* 3'UTR.

References

- Jambor, H., Brunel, C. & Ephrussi, A. Dimerization of *oskar* 3' UTRs promotes hitchhiking for RNA localization in the *Drosophila* oocyte. *Rna* **17**, 2049–2057 (2011).
- Lever, A., Gottlinger, H., Haseltine, W. & Sodroski, J. Identification of a sequence required for efficient packaging of human immunodeficiency virus type 1 RNA into virions. *J Virol* **63**, 4085–4087 (1989).
- Dubois, N., Marquet, R., Paillart, J.-C. & Bernacchi, S. Retroviral RNA dimerization: from structure to functions. *Front Microbiol* **9**, 527 (2018).
- Skripkin, E., Paillart, J. C., Marquet, R., Ehresmann, B. & Ehresmann, C. Identification of the primary site of the human immunodeficiency virus type 1 RNA dimerization in vitro. *Proc Natl Acad Sci U S A* **91**, (1994).
- Clever, J. L., Wong, M. L. & Parslow, T. G. Requirements for kissing-loop-mediated dimerization of human immunodeficiency virus RNA. *J Virol* **70**, (1996).
- Hussein, I. T. M. *et al.* Delineation of the Preferences and Requirements of the Human Immunodeficiency Virus Type 1 Dimerization Initiation Signal by Using an In Vivo Cell-Based Selection Approach. *J Virol* **84**, (2010).
- Elguindy, M. M. & Mendell, J. T. NORAD-induced Pumilio phase separation is required for genome stability. *Nature* **595**, 303–308 (2021).
- Bose, M., Lampe, M., Mahamid, J. & Ephrussi, A. Liquid-to-solid phase transition of *oskar* ribonucleoprotein granules is essential for their function in *Drosophila* embryonic development. *Cell* **185**, 1308-1324 e23 (2022).
- Little, S. C., Sinsimer, K. S., Lee, J. J., Wieschaus, E. F. & Gavis, E. R. Independent and coordinate trafficking of single *Drosophila* germ plasm mRNAs. *Nat Cell Biol* **17**, 558–568 (2015).

10. Mohr, S. *et al.* Opposing roles for Egalitarian and Staufen in transport, anchoring and localization of oskar mRNA in the Drosophila oocyte. *PLoS Genet* **17**, (2021).
11. Gáspár, I. *et al.* An RNA-based feed-forward mechanism ensures motor switching in oskar mRNA transport. *Journal of Cell Biology* **222**, (2023).
12. González-Reyes, A., Elliott, H. & St Johnston, D. Polarization of both major body axes in drosophila by gurken-torpedo signalling. *Nature* **375**, (1995).
13. Jain, A. & Vale, R. D. RNA phase transitions in repeat expansion disorders. *Nature* **546**, 243–247 (2017).
14. Van Treeck, B. & Parker, R. Emerging Roles for Intermolecular RNA-RNA Interactions in RNP Assemblies. *Cell* **174**, 791–802 (2018).
15. Poudyal, R. R., Sieg, J. P., Portz, B., Keating, C. D. & Bevilacqua, P. C. RNA sequence and structure control assembly and function of RNA condensates. *Rna* **27**, 1589–1601 (2021).
16. Tauber, D. *et al.* Modulation of RNA Condensation by the DEAD-Box Protein eIF4A. *Cell* **180**, 411-426 e16 (2020).
17. Garcia-Jove Navarro, M. *et al.* RNA is a critical element for the sizing and the composition of phase-separated RNA–protein condensates. *Nat Commun* **10**, (2019).
18. Mollieux, A. *et al.* Phase Separation by Low Complexity Domains Promotes Stress Granule Assembly and Drives Pathological Fibrillization. *Cell* **163**, (2015).
19. Langdon, E. M. *et al.* mRNA structure determines specificity of a polyQ-driven phase separation. *Science (1979)* **360**, 922–927 (2018).
20. Wang, S and Xu, Y. RNA structure promotes liquid-to-solid phase transition of short RNAs in neuronal dysfunction. *Communications Biology* **137**, (2024).
21. Smith, J. L., Wilson, J. E. & Macdonald, P. M. Overexpression of oskar directs ectopic activation of nanos and presumptive pole cell formation in Drosophila embryos. *Cell* **70**, (1992).
22. Ephrussi, A. & Lehmann, R. Induction of germ cell formation by oskar. *Nature* **358**, (1992).
23. Eichler, C. E., Hakes, A. C., Hull, B. & Gavis, E. R. Compartmentalized oskar degradation in the germ plasm safeguards germline development. *Elife* **9**, (2020).

Decision Letter, first revision:

Our ref: NCB-LE51679A

14th May 2024

Dear Dr. Ephrussi,

Thank you for submitting your revised manuscript "An architectural role of specific RNA-RNA interactions in oskar granules" (NCB-LE51679A). It has now been seen by the original referees and their comments are below. The reviewers find that the paper has improved in revision, and therefore we'll be happy in principle to publish it in Nature Cell Biology, pending minor revisions to satisfy the referees' final requests and to comply with our editorial and formatting guidelines.

The current version of your manuscript is in a PDF format, so please email us a copy of the file in an editable format (Microsoft Word or LaTeX)-- we can not proceed with PDFs at this stage.

Thank you again for your interest in Nature Cell Biology Please do not hesitate to contact me if you have any questions.

Sincerely,
Daryl

Daryl Jason Verzosa David, PhD

Senior Editor, Nature Cell Biology
Nature Portfolio
Advisory Editor, npj Biological Physics and Mechanics

Heidelberger Platz 3, 14197 Berlin, Germany
Email: daryl.david@nature.com
ORCID: <https://orcid.org/0000-0002-9253-4805>

Reviewer #1 (Remarks to the Author):

In the revised version of the manuscript, the authors have responded positively to my initial concerns

by performing additional experiments. The quality of the additional experiments is high enough to support the authors' arguments. Their updated results further reveal a much more complex mechanism of spatiotemporal control of oskar RNA localization and translation. The inclusion of the data on Egl in their pull-down assays indicates that, at least in vitro, a simple competition between Staufen and Egl for interaction with the SL2b stems would not explain the behavior of the oskar RNP. In addition, the finding that replacing the loop sequence (CCGCGG) with the HIV-2 sequence (GCGCGC) did not completely rescue RNA oligomerization and condensate formation in vitro is surprising; the authors' discussion (lines 371-379) is compelling. Although it would be interesting to know the difference between intact and the HIV-2 version of SL2b stem-loops in their 3D structures, this topic is far beyond the scope of this manuscript. Although several questions remain to be addressed, the authors adequately address my initial concerns and have already provided a nice piece of work that deepens our understanding of how a specific RNA-RNA interaction stimulates phase separation. In summary, I support the publication of this revised version in Nature Cell Biology.

Reviewer #2 (Remarks to the Author):

Overall I'm satisfied with the revisions, although it is a little bit unsatisfying that the authors were not able to conclusively demonstrate that kissing loop dimerization and not kissing loop-protein interactions underlie oskar granule assembly, but the HIV sequence swaps at least provide further support. Reversing the sequence would probably have been a better sequence swap but it could be difficult to conclusively interpret a negative result for any sequence. Perhaps at line 379 of the discussion it's worth mentioning that protein interactions with the kissing loop sequence motif could also contribute to granule formation for the sake of full transparency, but I would leave that up to the authors.

Also, I'm still a little bit confused about figure 2d. You state in the legend that there are three independent replicates, but 5 data points are shown. You note that $n = 5$ oocytes but it's not clear what the relationship between replicates and oocyte number is because I would think n in this context refers to replicates. Furthermore, the variation is just the standard deviation from the mean of these WT replicates which should be something that you can calculate even though it's a ratio rather than absolute value. If you're not able to calculate the variation in WT, the statistical analysis here is invalid.

Reviewer #3 (Remarks to the Author):

I appreciate the effort and attention to detail put into preparing this revised manuscript. I think it is ready for publication and makes a valuable and fascinating contribution to our understanding of RNA in condensates.

Decision Letter, final checks:

Our ref: NCB-LE51679A

13th June 2024

Dear Dr. Ephrussi,

Thank you for your patience as we've prepared the guidelines for final submission of your Nature Cell Biology manuscript, "An architectural role of specific RNA-RNA interactions in oskar granules" (NCB-LE51679A). Please carefully follow the step-by-step instructions provided in the attached file, and add a response in each row of the table to indicate the changes that you have made. Ensuring that each point is addressed will help to ensure that your revised manuscript can be swiftly handed over to our production team.

In recognition of the time and expertise our reviewers provide to Nature Cell Biology's editorial process, we would like to formally acknowledge their contribution to the external peer review of your manuscript entitled "An architectural role of specific RNA-RNA interactions in oskar granules". For those reviewers who give their assent, we will be publishing their names alongside the published article.

Nature Cell Biology offers a Transparent Peer Review option for new original research manuscripts submitted after December 1st, 2019. As part of this initiative, we encourage our authors to support increased transparency into the peer review process by agreeing to have the reviewer comments, author rebuttal letters, and editorial decision letters published as a Supplementary item. When you submit your final files please clearly state in your cover letter whether or not you would like to participate in this initiative. Please note that failure to state your preference will result in delays in accepting your manuscript for publication.

Cover suggestions

COVER ARTWORK: We welcome submissions of artwork for consideration for our cover. For more information, please see our guide for cover artwork.

Nature Cell Biology has now transitioned to a unified Rights Collection system which will allow our Author Services team to quickly and easily collect the rights and permissions required to publish your work. Approximately 10 days after your paper is formally accepted, you will receive an email in providing you with a link to complete the grant of rights. If your paper is eligible for Open Access, our Author Services team will also be in touch regarding any additional information that may be required to arrange payment for your article.

Please note that *Nature Cell Biology* is a Transformative Journal (TJ). Authors may publish their research with us through the traditional subscription access route or make their paper immediately open access through payment of an article-processing charge (APC). Authors will not be required to make a final decision about access to their article until it has been accepted. Find out more about Transformative Journals

Please use the following link for uploading these materials:
<https://mts-ncb.nature.com/cgi-bin/main.plex?el=A7C7vR6A2Bkfr6J3A9ftdgkbABEcANYM1Yd3xMOgg7QZ>

Best regards,

Kendra Donahue
Staff
Nature Cell Biology

On behalf of

Daryl Jason Verzosa David, PhD

Senior Editor, Nature Cell Biology
Advisory Editor, npj Biological Physics and Mechanics
Nature Portfolio

Heidelberger Platz 3, 14197 Berlin, Germany
Email: daryl.david@nature.com
ORCID: <https://orcid.org/0000-0002-9253-4805>

Reviewer #1:

Remarks to the Author:

In the revised version of the manuscript, the authors have responded positively to my initial concerns by performing additional experiments. The quality of the additional experiments is high enough to support the authors' arguments. Their updated results further reveal a much more complex mechanism of spatiotemporal control of oskar RNA localization and translation. The inclusion of the data on Egl in their pull-down assays indicates that, at least in vitro, a simple competition between Stauf and Egl for interaction with the SL2b stems would not explain the behavior of the oskar RNP. In addition, the finding that replacing the loop sequence (CCGCGG) with the HIV-2 sequence (GCGCGC) did not completely rescue RNA oligomerization and condensate formation in vitro is surprising; the authors' discussion (lines 371-379) is compelling. Although it would be interesting to know the difference between intact and the HIV-2 version of SL2b stem-loops in their 3D structures, this topic is far beyond the scope of this manuscript. Although several questions remain to be addressed, the authors adequately address my initial concerns and have already provided a nice piece of work that deepens our understanding of how a specific RNA-RNA interaction stimulates phase separation. In summary, I support the publication of

this revised version in Nature Cell Biology.

Reviewer #2:

Remarks to the Author:

Overall I'm satisfied with the revisions, although it is a little bit unsatisfying that the authors were not able to conclusively demonstrate that kissing loop dimerization and not kissing loop-protein interactions underlie oskar granule assembly, but the HIV sequence swaps at least provide further support. Reversing the sequence would probably have been a better sequence swap but it could be difficult to conclusively interpret a negative result for any sequence. Perhaps at line 379 of the discussion it's worth mentioning that protein interactions with the kissing loop sequence motif could also contribute to granule formation for the sake of full transparency, but I would leave that up to the authors.

Also, I'm still a little bit confused about figure 2d. You state in the legend that there are three independent replicates, but 5 data points are shown. You note that $n = 5$ oocytes but it's not clear what the relationship between replicates and oocyte number is because I would think n in this context refers to replicates. Furthermore, the variation is just the standard deviation from the mean of these WT replicates which should be something that you can calculate even though it's a ratio rather than absolute value. If you're not able to calculate the variation in WT, the statistical analysis here is invalid.

Reviewer #3:

Remarks to the Author:

I appreciate the effort and attention to detail put into preparing this revised manuscript. I think it is ready for publication and makes a valuable and fascinating contribution to our understanding of RNA in condensates.

Author Rebuttal, first revision:

Response to the Reviewers' comments

Reviewer #1 (Remarks to the Author):

In the revised version of the manuscript, the authors have responded positively to my initial concerns by performing additional experiments. The quality of the additional experiments is high enough to support the authors' arguments. Their updated results further reveal a much more complex mechanism of spatiotemporal control of oskar RNA localization and translation. The inclusion of the data on Egl in their pull-down assays indicates that, at least in vitro, a simple competition between Staufen and Egl for interaction with the SL2b stems would not explain the behavior of the oskar RNP. In addition, the finding that replacing the loop sequence (CCGCGG) with the HIV-2 sequence (GCGCGC) did not completely rescue RNA oligomerization and condensate formation in vitro is surprising; the authors' discussion (lines 371-379) is compelling. Although it would be interesting to know the difference between intact and the HIV-2 version of SL2b stem-loops in their 3D structures, this topic is far beyond the scope of this manuscript. Although several questions remain to be addressed, the authors adequately address my initial concerns and have already provided a nice piece of work that deepens our understanding of how a specific RNA-RNA interaction stimulates phase separation. In summary, I support the publication of this revised version in Nature Cell Biology.

We are pleased that the reviewer appreciates our additional experiments and support publication.

Reviewer #2 (Remarks to the Author):

Overall I'm satisfied with the revisions, although it is a little bit unsatisfying that the authors were not able to conclusively demonstrate that kissing loop dimerization and not kissing loop-protein interactions underlie oskar granule assembly, but the HIV sequence swaps at least provide further support. Reversing the sequence would probably have been a better sequence swap but it could be difficult to conclusively interpret a negative result for any sequence. Perhaps at line 379 of the discussion it's worth mentioning that protein interactions with the kissing loop sequence motif could also contribute to granule formation for the sake of full transparency, but I would leave that up to the authors.

Also, I'm still a little bit confused about figure 2d. You state in the legend that there are three independent replicates, but 5 data points are shown. You note that $n = 5$ oocytes but it's not clear what the relationship between replicates and oocyte number is because I would think n in this context refers to replicates. Furthermore, the variation is just the standard deviation from the mean of these WT replicates which should be something that you can calculate even though it's a ratio rather than absolute value. If you're not able to calculate the variation in WT, the statistical analysis here is invalid.

Following the reviewer's suggestion, we have now included a sentence in the discussion (lines 379-382 in the main text).

For the revision, as suggested by Reviewer 1 (minor comments > point no.3), we performed two additional replicates of the RNA affinity capture experiment and included the data in the quantification (shown in Figure 2d). However, in the corresponding legend we by mistake did not change the number of replicates from 'three' to 'five', which has now been pointed out by

the reviewer. Importantly, n=5 indicates 5 independent biological replicates and not 5 oocytes as rightly pointed out by the reviewer. We apologize for the confusion and have corrected the errors in the attached main text file.

Also, regarding the variation within the WT in Figure 2d, we would like to clarify once again that the plotted standard deviation does not reflect the inherent variation within one sample type. For each replicate, we first quantified the relative levels of the WT and UU RNAs (setting WT to 1). The obtained factor was then used to normalize the amount of Bruno pulled down by UU. Therefore, these are not absolute values and the error bar on UU represents the variation in 'relative enrichment' of Bruno between WT and UU in the different experimental replicates. Since the data depict five independent biological replicates, the intrinsic variation within one sample type is not relevant.

Reviewer #3 (Remarks to the Author):

I appreciate the effort and attention to detail put into preparing this revised manuscript. I think it is ready for publication and makes a valuable and fascinating contribution to our understanding of RNA in condensates.

We are happy that the reviewer finds the revisions satisfactory and appreciates the contribution of the study in the field of condensate biology.

Final Decision Letter:

Dear Dr Ephrussi,

I am pleased to inform you that your manuscript, "An architectural role of specific RNA-RNA interactions in oskar granules", has now been accepted for publication in Nature Cell Biology.

Please note that *Nature Cell Biology* is a Transformative Journal (TJ). Authors may publish their research with us through the traditional subscription access route or make their paper immediately open access through payment of an article-processing charge (APC). Authors will not be required to make a final decision about access to their article until it has been accepted. Find out more about Transformative Journals

If you have not already done so, we strongly recommend that you upload the step-by-step protocols used in this manuscript to protocols.io (<https://protocols.io>), an open online resource that allows

researchers to share their detailed experimental know-how. All uploaded protocols are made freely available and are assigned DOIs for ease of citation. Protocols and Nature Portfolio journal papers in which they are used can be linked to one another, and this link is clearly and prominently visible in the online versions of both. Authors who performed the specific experiments can act as primary authors for the Protocol as they will be best placed to share the methodology details, but the Corresponding Author of the present research paper should be included as one of the authors. By uploading your Protocols onto protocols.io, you are enabling researchers to more readily reproduce or adapt the methodology you use, as well as increasing the visibility of your protocols and papers. You can also establish a dedicated workspace to collect your lab Protocols. Further information can be found at <https://www.protocols.io/help/publish-articles>.

With kind regards,

Daryl

Daryl Jason Verzosa David, PhD

Senior Editor, Nature Cell Biology
Advisory Editor, npj Biological Physics and Mechanics
Nature Portfolio

Heidelberger Platz 3, 14197 Berlin, Germany
Email: daryl.david@nature.com
ORCID: <https://orcid.org/0000-0002-9253-4805>